# Human CD4 cytotoxic T lymphocytes mediate potent tumor control in humanized immune system mice

Wen Lin[1], Varan Singh[1], Raynel Springer[1], Gabrielle Choonoo[1], Namita Gupta[1], Aditi Patel[1], Davor Frleta [1], Jun Zhong[1], Tomasz Owczarek[1], Corinne Decker[1], Lynn Macdonald[1], Andrew Murphy [1], Gavin Thurston [1], Markus Mohrs[1], Ella Ioffe[1] & Yi-Fen Lu [1✉]

Efficacy of immune checkpoint inhibitors in cancers can be limited by CD8 T cell dysfunction or HLA-I down-regulation. Tumor control mechanisms independent of CD8/HLA-I axis would overcome these limitations. Here, we report potent CD4 T cell-mediated tumor regression and memory responses in humanized immune system (HIS) mice implanted with HT-29 colorectal tumors. The regressing tumors showed increased CD4 cytotoxic T lymphocyte (CTL) infiltration and enhanced tumor HLA-II expression compared to progressing tumors. The intratumoral CD4 T cell subset associated with tumor regression expressed multiple cytotoxic markers and exhibited clonal expansion. Notably, tumor control was abrogated by depletion of CD4 but not CD8 T cells. CD4 T cells derived from tumor-regressing mice exhibited HLA-II-dependent and tumor-specific killing ex vivo. Taken together, our study demonstrates a critical role of human CD4 CTLs in mediating tumor clearance independent of CD8 T cells and provides a platform to study human anti-tumor immunity in vivo.

[1] Regeneron Pharmaceuticals, 777 Old Saw Mill River Road, Tarrytown, NY 10591, USA. ✉email: yifen.lu@gmail.com

mmune checkpoint inhibitors (ICIs) exhibit robust and durable anti-tumor responses in some cancer patients and have revolutionized treatment for multiple cancer types. However, a sizable proportion of patients show resistance to the therapy[1,2]. ICIs boost the abundance and function of CD8 cytotoxic T lymphocytes (CTLs), which recognize tumor antigens presented by HLA class I (HLA-I) and mediate tumor-specific killing[3,4]. Dysfunction of CD8 CTLs and/or loss of HLA-I by tumor cells appear to be the most common mechanisms of ICI resistance[5–7]. Therefore, it is important to identify tumor control mechanisms operating independently of the CD8/HLA-I axis to improve patient outcomes.

CD4 T helper cells (Th) are important for inducing effective CD8 CTL responses by providing activation signals to antigen-presenting cells (APCs)[4,8]. CD4 T cells recognize antigens presented by HLA class II (HLA-II), which are traditionally expressed by APCs, but are increasingly detected in a variety of cancer cells[9]. Multiple lines of clinical evidence have revealed an important role of the CD4/HLA-II axis in tumor control. Tumor HLA-II expression is associated with improved prognosis and response to anti-PD1/PD-L1 therapies, as well as an increase in tumor-infiltrating lymphocytes (TILs) and interferon signaling[10–14]. Tumor-antigen-specific CD4 T cells are frequently detected in cancer patients[15–17], and vaccination that boosts tumor-antigen-specific CD4 and CD8 T cells has resulted in clinical benefit including cases of complete response[18–20]. Adoptive transfer of tumor-antigen-specific CD4 T cells has been reported to potently control tumor progression[21–23]. Although the anti-tumor activity of the CD4/HLA-II axis has generally been attributed to its helper function in boosting CD8 CTLs, direct tumor-targeting by CD4 T cells has not been examined closely.

CD4 T cells containing cytotoxic granules and exhibiting ex vivo cytolytic activity (CD4 CTLs) have been detected in cancer patients. Single-cell RNA (scRNAseq) profiling of colorectal tumor specimens has revealed a Th1-like CD4 cluster that expresses both Th1 and cytotoxic markers. The enrichment of this cluster in microsatellite-instable tumors (MSI) was speculated to underlie the favorable responses of this tumor type to ICI[24]. Similarly, a study of T cell subsets in bladder tumors has identified two clonally expanded CD4 CTL clusters expressing a core set of cytotoxic molecules. CD4 CTLs can kill autologous tumor cells in an HLA-II-restricted manner ex vivo, and a CD4 CTL gene signature predicts response to anti-PD-L1 therapy in patients with inflamed tumors[25]. More recently, CD4 CTL clusters have also been identified in melanoma and other tumor types. CD4 CTL-mediated killing of tumor cells ex vivo is enhanced by a SLAMF7 agonist[26]. Collectively, these studies demonstrate that human CD4 CTLs possess cytolytic activity that can be boosted by immunotherapies ex vivo. However, their independent contribution to tumor control in vivo has not been demonstrated.

Mouse tumor studies have provided important insights into tumor-specific CD4 CTL functions. Mouse CD4 CTLs have been described predominantly in adoptive transfer models and occasionally syngeneic tumor models following CTLA4 blockade, OX40 costimulation, 4-1BB costimulation combined with a tumor vaccine, or tumor-targeted expression of highly immunogenic tetanus toxoid protein (TT)[27–32]. Adoptively transferred tumor-antigen-specific mouse CD4 T cells are sufficient to elicit potent tumor control in lymphopenic or Rag KO mice. This tumor control requires IFNγ production by CD4 T cells, MHC-II expression on tumor cells, and is independent of host immune cells[27,28]. It has recently been shown that tumor-targeted TT protein expression induces TT-specific mouse CD4 CTLs, which are required for robust tumor control[32]. Furthermore, mouse CD4 CTL differentiation and function were shown to require IL-2

and were suppressed by Tregs limiting IL-2 availability[33]. Despite insights gained from mouse studies, the relevance to human CD4 CTL biology remains to be examined.

Investigating human CD4 CTL biology in vivo has been hampered by the lack of suitable preclinical models. Humanized immune system (HIS) mice harboring human donor stem-cell-derived immune cells allow a unique opportunity to tackle this question[34,35]. Human T cells developed in HIS mice tolerate both host MHC and donor HLA[36,37] and closely recapitulate several key aspects of human T cell biology, including expression of major lineage and functional markers, as well as responsiveness to immunotherapies in some tumor models[38–40]. In the current study, we describe a novel HIS mouse tumor model that shows potent tumor control and memory responses. This tumor control required CD4, but not CD8 T cells, thereby allowing us to assess the role of CD4 T cells in tumor control independent of their helper function. Increased HLA-II expression on tumor cells, as well as increased frequencies of CD4 CTLs in the blood and tumors, are consistent with the direct tumor-killing function of CD4 CTLs. scRNAseq profiling further confirmed an association between tumor control and an intratumoral CD4 CTL subset exhibiting effector phenotype and clonal expansion. CD4 T cells derived from regressing tumors exhibited tumor-specific HLA-II-restricted tumor cell killing ex vivo. Collectively, our study provides strong evidence that human CD4 CTLs are capable of killing tumor cells directly and controlling tumor growth independent of CD8 T cells. Additionally, the described HIS tumor model provides a valuable preclinical platform to interrogate human CD4 CTL biology and establish therapeutic strategies to enhance their differentiation and function.

## Results

**HIS mice show potent tumor control and memory responses against HT-29.** To examine the anti-tumor activity of human immune cells in vivo, HIS mice were generated and subsequently challenged with human tumor cells implanted subcutaneously (Fig. 1A). HIS mice showed robust reconstitution with multiple human immune cell lineages (Supplementary Fig. 1A). Without treatment, most human solid tumor cell lines commonly show consistent progression when implanted into HIS mice. However, human immune cells could mediate spontaneous tumor regression of HT-29, a human colorectal cancer cell line, offering an opportunity to examine the mechanism of tumor clearance. Of 223 HIS mice engrafted with human CD34+ hematopoietic stem and progenitor cells (HSPCs) from 27 different donors and implanted with HT-29 tumors, 76 (34%) rejected the tumors, 40 (18%) showed tumor regression, and 107 (48%) showed tumor progression (Fig. 1A). Since tumor regression is a dynamic process preceding rejection, the combined regression and rejection groups were designated the tumor-regressing group (R) and compared to the tumor-progressing group (P) throughout the study. Tumor regression required human immune cells, as nearly all non-reconstituted mice showed tumor progression (Fig. 1A). Spontaneous tumor regression/rejection was observed in both male and female mice and appeared more common in the female group ($p < 0.0001$, Chi-square test) (Supplementary Fig. 1B). Similar to a previous report[38], tumor growth outcome was not correlated with the degree of HLA compatibility between human donors and HT-29 cells (Supplementary Fig. 1C).

To examine anti-tumor memory responses, 37 mice that had previously rejected the primary tumors were rechallenged with HT-29 cells at least 60 days after tumor clearance (Fig. 1A). 10 (19%) of the rechallenged mice remained tumor-free. Additionally, tumor rechallenge resulted in a higher rejection rate (54% vs. 34%) and lower progression rate (27% vs. 48%)

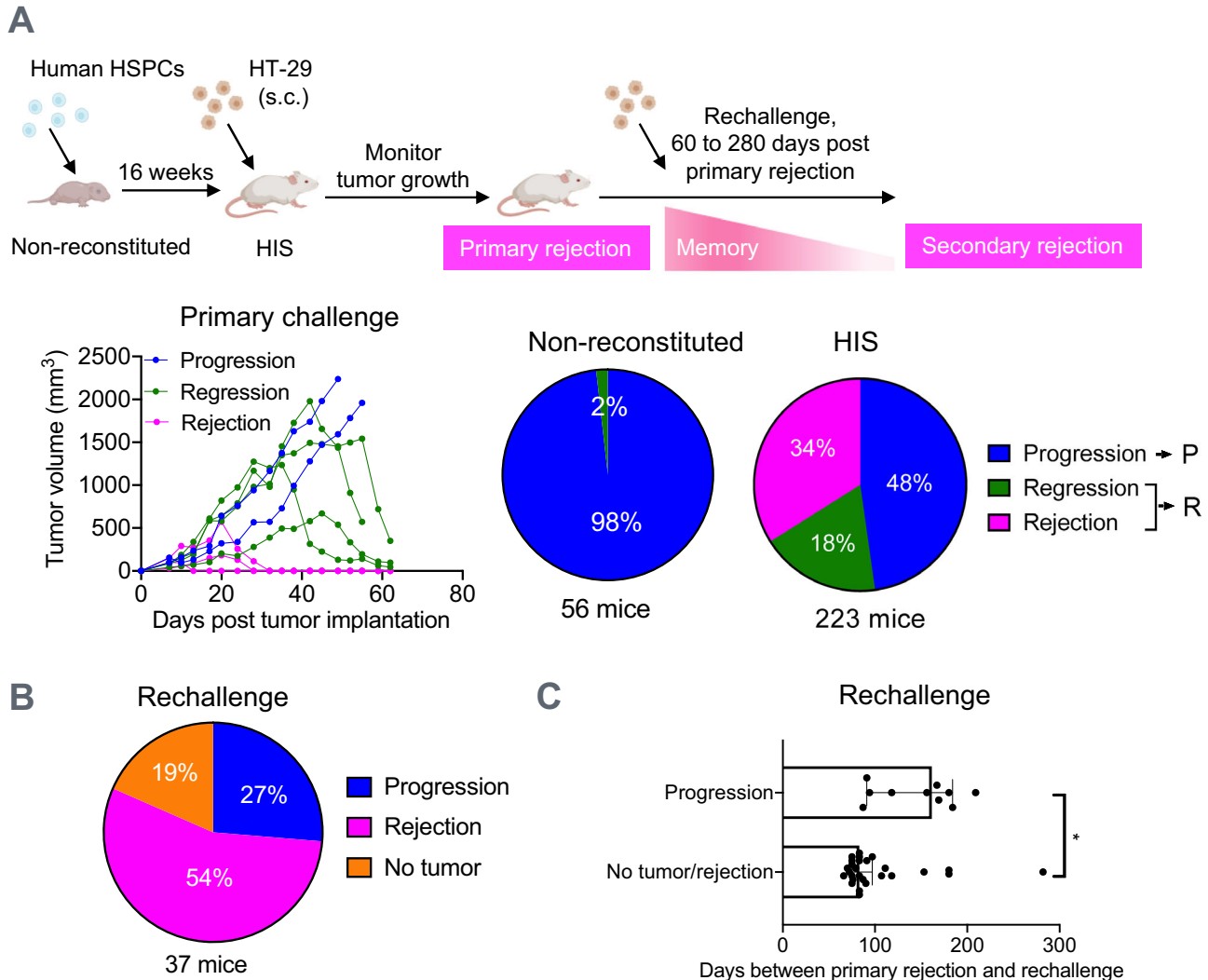

**Fig. 1 HIS mice show potent tumor growth control and memory responses against HT-29. A** Schematic of HIS tumor studies (created with BioRender.com), tumor growth curve of a representative cohort, and summarized tumor growth outcome of 56 non-reconstituted and 223 HIS mice following the primary HT-29 tumor challenge. Tumor growth curve is generated using repeated measurements of the same mouse overtime. Summarized tumor growth outcome results are shown as pie charts. The numbers in the pie charts represent the fraction of mice for each tumor growth outcome. The 223 HIS mice were reconstituted with HSPCs from 27 human donors. Regression is defined as having two tumor volume values smaller than the preceding peak value. Progression group: P; combined regression and rejection groups: R. **B** Mice that rejected HT-29 tumors during the primary challenge were rechallenged with HT-29 cells. The 37 mice were reconstituted with HSPCs from 10 human donors. **C** Comparison of the time span between primary tumor rejection and rechallenge tumor growth outcome. Results of tumor progression ($n = 10$) and no tumor/rejection ($n = 27$) are shown as median ± 95% confidence interval. *$p < 0.05$, unpaired $t$ test.

compared to the primary challenge (Fig. 1B), revealing anti-tumor memory that protected mice from the secondary tumor challenge. Notably, mice that cleared secondary tumors were mostly rechallenged around 80 days since they cleared the primary tumors; whereas mice that failed to control secondary tumors were often rechallenged around 160 days (median 83 vs. 162 days, $p = 0.01$) (Fig. 1C). This suggests that protective memory lasts for at least 80 days but wanes over time.

**CD4, but not CD8 T cells are required for tumor control.** Similar to humans, HIS mice inherently exhibit variable immune profiles. To explore the immunological differences that may predict a favorable tumor growth outcome, we profiled human immune cell reconstitution and activation phenotypes in 186 mice prior to tumor implantation and correlated these parameters to tumor growth outcome. Mice bearing regressing tumors showed significantly increased numbers of human CD45

leukocytes and CD3 T cells in circulation, of which both conventional CD4 (CD4 Tconv) and CD8 T cells were higher than those in mice bearing progressing tumors (Fig. 2A and Supplementary Fig. 2A). The frequencies and ratios of CD4 and CD8 out of total T cells were similar between the two groups (Supplementary Fig. 2B). Tumor regression was associated with an enrichment of naive CD4, but not naive CD8 T cells (Fig. 2B). In addition, mice that rejected tumors showed a significantly reduced Treg frequency and increased T effector/Treg ratio compared to mice with tumor progression (Fig. 2C). These observations imply that both CD4 Tconv and CD8 T cells could participate in tumor control.

To dissect the individual contribution of CD4 and CD8 T cells to tumor control, we depleted each T cell subset prior to tumor implantation and examined the effects on tumor growth. Analysis of circulating T cells after tumor implantation demonstrated high efficiency of depletion (Supplementary Fig. 2C). Notably, tumor

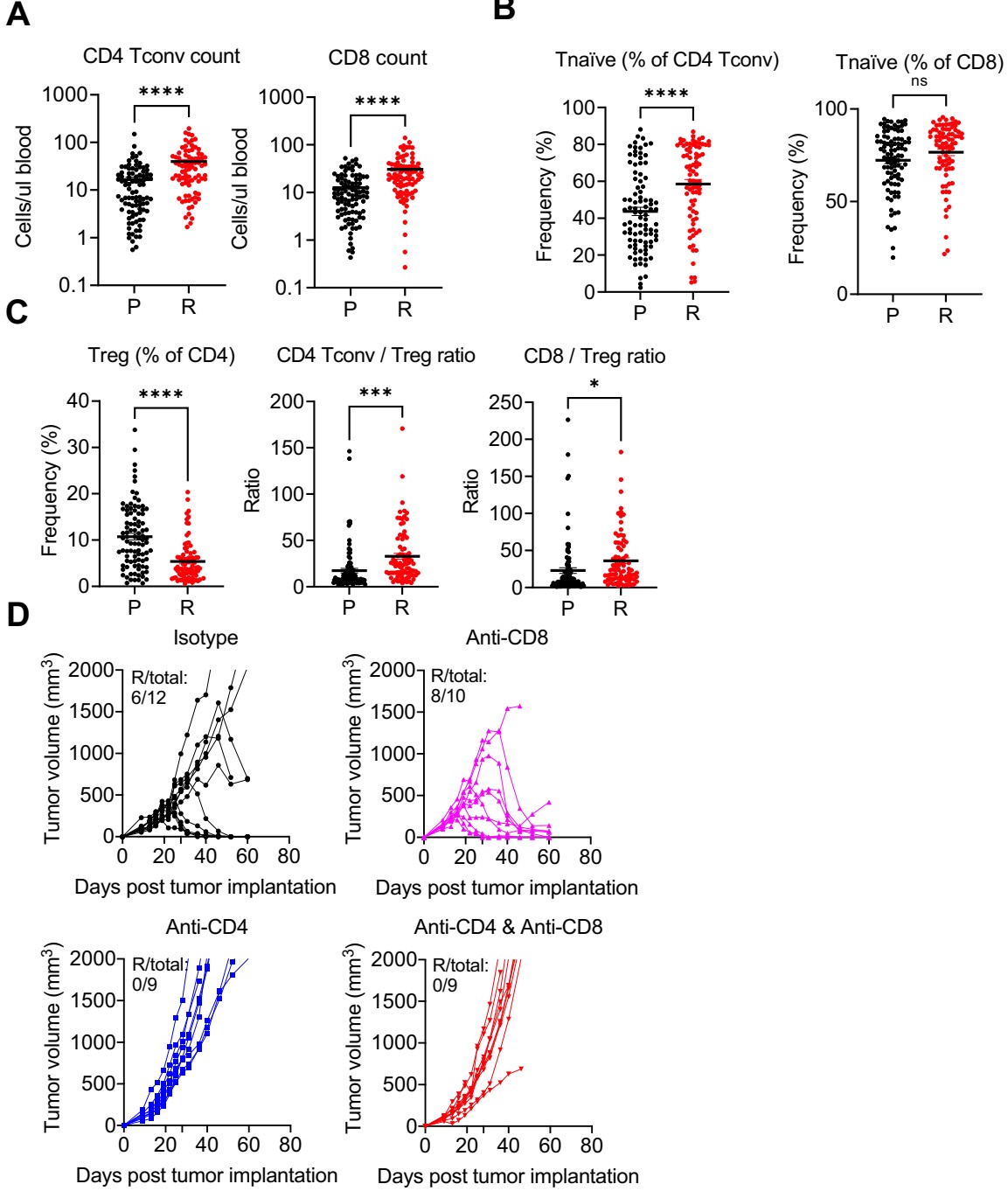

**Fig. 2 CD4, but not CD8 T cells are required for tumor control. A–C** Association between tumor growth outcome and T-cell phenotypes in the circulation before tumor implantation. CD4 Tconv: CD4+FOXP3−; Treg: CD4+FOXP3+; T naïve: CD45RA+CCR7+. Association from P ($n = 101$) and R ($n = 83$) are shown. The results are shown as means ± SEM. *$p < 0.05$, **$p < 0.01$, ****$p < 0.0001$, unpaired $t$ test. D Tumor growth curve of mice in the control (isotype), CD8-depleted (anti-CD8), CD4-depleted (anti-CD4), and CD4 & CD8-depleted groups (anti-CD4 & anti-CD8). The number of mice that show tumor regression or rejection out of the total number of mice of each group is shown. Antibody treatments started 9 days before tumor implantation and lasted throughout the study. Tumor growth curve is generated using repeated measurements of the same mouse over time.

control was largely retained in CD8-depleted mice but was completely abolished upon CD4 or combined CD4 and CD8 depletion (Fig. 2D). It is worth noting that depletion of CD8 T cells in our model resulted in a somewhat higher rate of tumor rejection. It is plausible that CD8 depletion decreases the consumption of T cell-supporting cytokines in HIS mice, leading to CD4 T cell expansion and favorable tumor control[41]. Taken together, these results demonstrates that CD4, but not CD8 T cells are required to control tumor growth.

**Tumor control is associated with HLA-II expression on HT-29 tumors**. Since CD4 T cells recognize foreign peptides presented by HLA-II, we examined HLA-II expression on HT-29 cells. In vitro, HT-29 cells did not express HLA-II at baseline, but HLA-II expression was upregulated in the presence of IFNγ (Fig. 3A). In vivo, EpCAM-gated HT-29 tumor cells harvested from non-reconstituted mice did not express HLA-II, whereas HT-29 tumor cells from HIS mice showed HLA-II expression (0.1% vs. 6% respectively, $p = 0.03$) (Fig. 3B). We also noted a positive

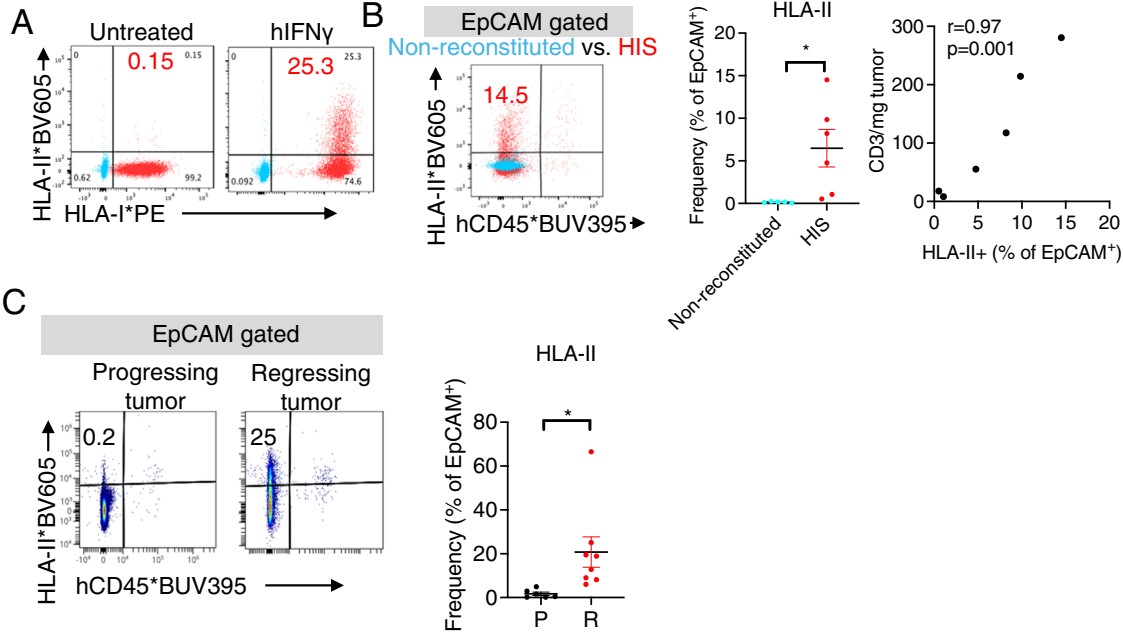

**Fig. 3 Tumor control is associated with HLA-II expression on HT-29 tumors. A** HLA-I and II expression on cultured HT-29 tumor cells treated with human IFNγ at 10 ng/ml for 16 hours. Untreated cells were used as control. **B** HLA-II expression on HT-29 tumors developed in non-reconstituted ($n = 5$) or HIS mice ($n = 6$) (left) and correlation between tumor HLA-II expression and CD3 cell density of HIS mice (right). The left graph is shown as means ± SEM. *$p < 0.05$, unpaired $t$ test. Pearson correlation coefficient and $p$ value are shown on the right graph. **C** HLA-II expression levels on progressing ($n = 6$) and regressing ($n = 8$) tumors. The result is shown as means ± SEM. *$p < 0.05$, unpaired $t$ test.

correlation between the level of HLA-II expression on tumor cells and the number of tumor-infiltrating T cells (Fig. 3B). Furthermore, EpCAM-gated tumor cells from regressing tumors showed increased HLA-II expression compared to progressing tumors (21% vs. 2% respectively, $p = 0.03$) (Fig. 3C). Collectively, these results suggest that IFNγ produced by tumor-infiltrating T cells induced HLA-II expression on HT-29 cells required for tumor recognition by CD4 CTLs.

To address whether CD4 T cell rejection of HT-29 is alloantigen-specific and better define the potential alloantigen response, we exploited the use of (1) CA46 cells, a B-cell lymphoma cell line expressing fully mismatched HLA-II to HT-29 and (2) HCT116 cells, another colorectal tumor cell line expressing mostly mismatched HLA-II except for DPA1. The HLA information of HT-29, CA46, and HCT116 cells is provided in Supplementary Data 1. In mice that had previously rejected HT-29 tumors, CA46 tumor implantation led to a heterogeneous tumor outcome, in which some mice developed large tumors quickly and others had lower tumor burden compared to control mice that had never experienced HT-29 tumors (Supplementary Fig. 3). These observations suggest that primary HT-29 rejection provided little or partial protection against CA46 tumor challenge in vivo. It is possible that some HT-29-recognizing TCRs cross-react to CA46 allo-antigens, as it has been reported that CD4 TCRs can promiscuously recognize different HLA-II alleles with shared epitopes[42–44]. Notably, primary HT-29 tumor rejection resulted in robust eradication of HCT116 tumors (Supplementary Fig 3). This result supports the possibility that CD4 T cells reject HT-29 tumors through DPA1 allo-recognition.

**Tumor control is associated with increased CD4 CTLs in blood and tumors.** To identify the immunological changes associated with a favorable tumor growth outcome, we used flow cytometry to profile human immune cells in blood and tumors. Mice with regressing tumors showed a significant increase in CD3 T cells in blood and tumor infiltrates (Supplementary Fig. 4A, B). Given the

critical role of CD4 T cells in tumor control and their potential tumor-killing capacity, we further characterized these cells using a panel of lineage and functional markers. Dimensionality reduction analysis revealed that CD4 T cells clustered primarily by the tissue of origin (blood vs. tumor) and tumor growth outcome (progression vs. regression) (Fig. 4A). Unsupervised clustering analysis identified 11 CD4 subsets exhibiting varying degrees of activation/dysfunction marker expression (Fig. 4B, Supplementary Fig. 4C, D).

In blood, mice that rejected tumors showed decreased naive and central memory T cells [flow cluster (FC)1-T naive/Tcm] and increased PD1^Low effector memory CD4 T cells (FC2-Tem-PD1^Low), suggesting more robust T-cell activation compared to mice with tumor progression (Supplementary Fig. 4C, E). Notably, tumor regression also associated with a significantly increased CD4 subset expressing Granzyme K (FC3-GZMK) (Fig. 4C). These immunological changes in the blood of tumor-regressing mice may represent biomarkers to monitor effective anti-tumor CD4 CTL immunity in circulation.

In tumors, three clusters of CD4 T cells—FC5-Tex, FC6-CXCL13, and FC7-GZMB & GNLY—expressed PD1 and another immune checkpoint (Fig. 4B and Supplementary Fig. 4C) molecules. The frequency of the FC5, 6, and 7 combined population was similar between progressing and regressing tumors, possibly because these clusters represent cells of different functional states. FC5 cells expressed high levels of Ki67 (Supplementary Fig. 4D), which has been associated with exhausted T cells with impaired anti-tumor activity[45]. Consistently, this population was enriched in progressing tumors (Supplementary Fig. 4E). FC6 cells highly expressed CXCL13 protein (Supplementary 4D), a chemokine associated with Th1, Tfh, and exhausted CD8 T cells in patient tumors[46–48]. Therefore, FC6 may represent cells with Th1 or Tfh function. The frequency of this cluster was similar between progressing and regressing tumors. FC7 cells expressed well-established cytotoxic markers GZMB and GNLY (Fig. 4B and Supplementary Fig. 4E) and were

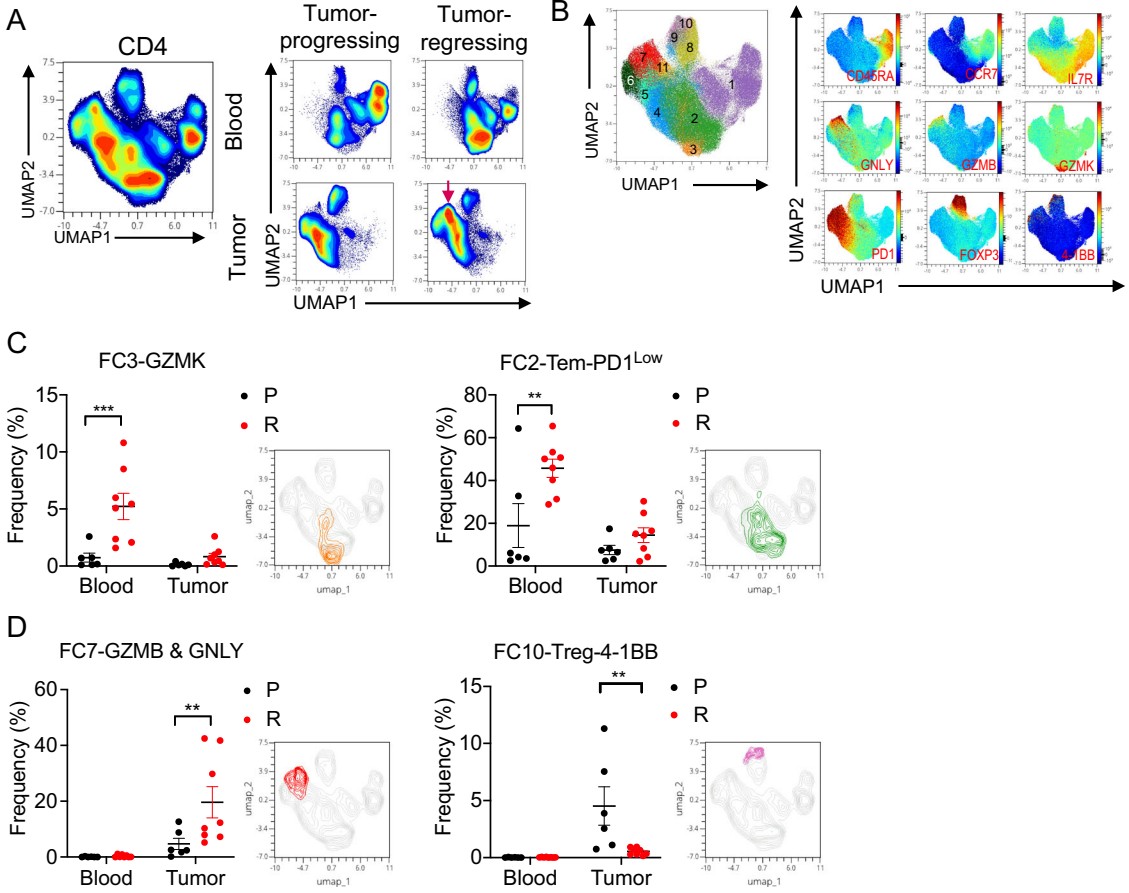

**Fig. 4 Tumor control is associated with increased CD4 CTLs in blood and tumors. A** Flow cytometry and dimensionality reduction analysis of CD4 T cells from blood and tumors of mice with tumor progression ($n = 6$) and regression ($n = 8$). The red arrow indicates a population enriched in regressing tumors. **B** Expression of selected T-cell lineage and functional markers across CD4 clusters. **C** Frequencies of clusters that are increased in the blood of tumor-regressing mice. **D** Frequencies of clusters that are increased in regressing tumors. The GZMK cluster is defined by GZMK expression. The GZMB & GNLY cluster is defined by GZMB and GNLY expression. The Tem-PD1$^{Low}$ cluster is defined by CD45RA$^-$CCR7$^-$IL7R$^+$PD1$^{Low}$. The results are shown as means ± SEM. **p < 0.01, ***p < 0.001, two-way ANOVA, Sidak's multiple comparison test.

enriched in regressing tumors (Fig. 4D). This CD4 cell cluster may represent the CD4 CTLs that mediate tumor control via direct cytotoxicity[25,26].

In addition, the suppressive Treg cluster (FC10-Treg-4-1BB)[49] was notably increased in progressing tumors, while tumor-progressing mice also showed increased Treg reconstitution prior to tumor implantation (Figs. 4D and 2C). Our data suggest that human Treg cells may suppress CD4-mediated tumor cell killing, consistent with previous reports delineating the interplay between Tregs and CD4 CTLs[25,33].

Parallel analysis of CD8 T cells revealed 9 clusters (Supplementary Fig. 4F). Compared to mice with progressing tumors, mice with regressing tumors showed an increased frequency of CD8 T cells expressing cytotoxic molecules in the blood (FC9-GNLY & GZMB & GZMK), but no increase in the frequency of cytotoxic CD8 T cells in tumors (FC7-GZMB and FC8-GNLY & GZMB) (Supplementary Fig. 4G). These findings are consistent with our previous observation that CD8 T cells were dispensable for controlling HT-29 tumor growth.

**Tumor control is associated with intratumoral PRF1$^+$ CD4 CTL subset.** To further refine the correlations between human immune cell phenotypes and tumor growth outcome, we profiled intratumoral human immune cells (hCD45$^+$) by scRNAseq and T-cell receptor sequencing (TCRseq). A total of 39,466 hCD45$^+$

cells were sorted from six progressing and five regressing tumors. After excluding potential dead cells, doublets, and mouse cells, 39,189 remaining cells were used for analysis. To capture the biological variability of the single cells, we limited the analysis to 2000 genes exhibiting the highest cell-to-cell variation. Specific immune cell subsets were annotated by expression of lineage and functional genes ($p < 0.01$, fold change >1.5). Differentially upregulated and representative gene sets of each CD4 cluster are listed in Supplementary Data 2 and Supplementary Table 1, respectively. Expression of key T cell lineage and functional markers on CD4 T cell clusters are shown in Fig. 5D, E, and Supplementary Fig. 5D. Unsupervised clustering of hCD45$^+$ infiltrates identified a diverse range of immune cell subsets, including T, B, myeloid, and NK cells (Supplementary Fig. 5A, B). Regressing tumors showed decreased pDC, proliferating Treg, DC, monocyte/macrophage, NK, and increased non-proliferating CD8 T cells (Supplementary Fig. 5C). Further analysis of CD4 T cells revealed nine subsets (Fig. 5A), of which two were enriched in regressing tumors and four were enriched in progressing tumors (Fig. 5B, C). Canonical CD4 subsets were readily detectable, including C5-Tnaive/Tcm and C6-Prolif. Three Treg subsets were detected, including C7-Treg, C8-Treg-TNFRSF9, and C9-Treg-Prolif. Compared with Tregs (C7), TNFRSF9 Tregs (C8) expressed higher levels of *IL2R* and co-stimulatory/coinhibitory receptors *TNFRSF4, TNFRSF18, TIGIT,* and *CTLA4* (Supplementary Data 2). In our study, all three Treg subsets were

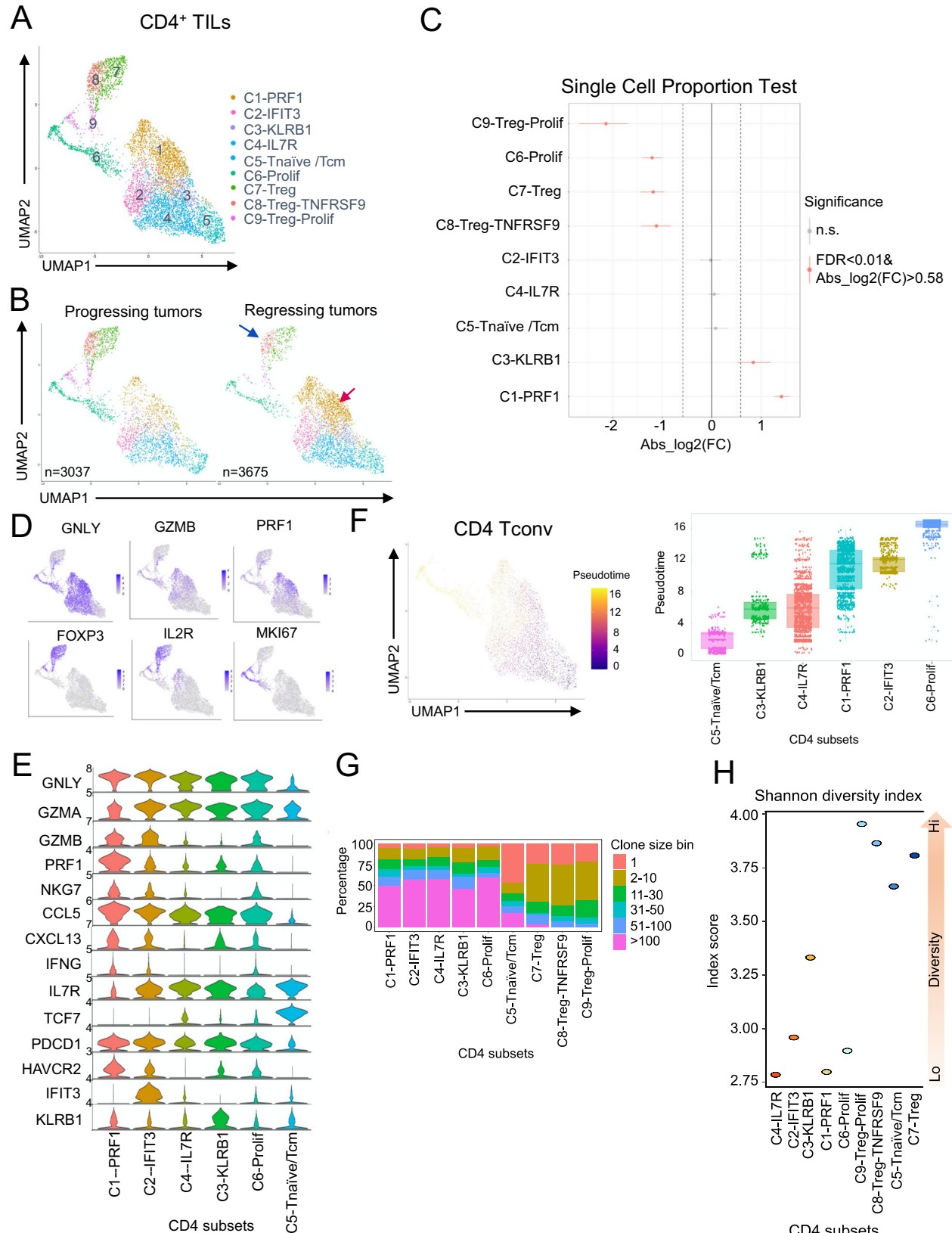

enriched in progressing tumors, indicating that human Tregs in HT-29 tumor microenvironment are immune-suppressive. (Fig. 5B, C).

Three CD4 subsets with cytotoxic features were identified, including C1-PRF1, C2-IFIT3, and C4-IL7R. Cells in these three clusters upregulated at least one cytotoxic marker (*GZMA*,

*GZMB*, *GNLY*, *NKG7*, and *PRF1*) (Supplementary Data 2). Cells in cluster C4-IL7R upregulated IL7R, a marker of CD4 CTL precursors[50]. C2-IFIT3 cells upregulated multiple interferon-induced genes, including *IFIT3* as the most distinctive one (Fig. 5E). Notably, C1-PRF1 demonstrated the strongest effector phenotype, reflected by high expression of cytotoxic (*PRF1* and

**Fig. 5 Tumor control is associated with intratumoral PRF1 + CD4 CTL subset expressing high levels of cytotoxic markers. A** Clusters of CD4 TILs identified by scRNAseq. **B** Clusters of CD4 TILs from progressing ($n = 3037$) and regressing ($n = 3675$) tumors. Blue and red arrows indicate markedly decreased and increased clusters, respectively, in regressing tumors. **C** Single-Cell Proportion Test comparison of CD4 TILs from progressing and regressing tumors. A significant difference is defined by false discovery rate (FDR) < 0.01 and the fold change of regressing over progressing tumors >1.5 or <−1.5. The dashed line indicates $y = 0.58$ (equivalent to the 1.5-fold change). **D** UMAP of selected T-cell lineage and functional markers, including CD4 CTL (GNLY, GZMB, and PRF1), Treg (FOXP3 and IL2R), and proliferation (MKI67). **E** Violin plots of selected T-cell lineage and functional markers, including CD4 CTL (GNLY, GZMA, GZMB, PRF1, NKG7, and CCL5), Th1 (CXCL13 and IFNG), T-naive/Tcm (IL7R and TCF7), activation/dysfunction (PDCD1 and HAVCR2), IFN-stimulated gene (IFIT3), and Th17 (KLRB1). **F** Pseudotime trajectory analysis of CD4 Tconv. **(G-H)** Percentage of TCRs of different clone size bins and Shannon diversity index of each CD4 cluster.

*NKG7*) and activation/dysfunctional markers (*PD1, HAVCR2/TIM3*), as well as low expression of Tnaive/Tcm markers (*TCF7* and *IL7R*) (Fig. 5D, E). The PRF1 cluster also upregulated Th1 markers *IFNG* and *CXCL13*, which contribute to tumor control by up-regulating HLA-II and promoting the formation of tertiary lymphoid structures (TLS), respectively[51,52] (Fig. 5E).

To examine the differentiation state of CD4 Tconv clusters, we performed pseudotime trajectory analysis. Genes associated with effector T cell function, including *GZMB* and *PRF1*, were upregulated along pseudotime axis, whereas genes associated with Tnaive/Tcm cells, including *TCF7* and *IL7R*, were downregulated (Supplementary Fig. 5E). The high pseudotime value of cells in the PRF1 cluster indicated high degree of differentiation, which was consistent with the effector cell state of this cluster (Fig. 5F). The C1-PRF1 cell cluster was highly enriched in tumor-regressing mice (Fig. 5B, C), consistent with our hypothesis that CD4 CTLs potently control tumors by direct tumor cell killing.

Another CD4 T cell cluster enriched in regressing tumors was C3-KLRB1 (Fig. 5C), which expressed several cytotoxic molecules and Th17 markers including *KLRB1, IL17A,* and *CCR6* (Fig. 5E and Supplementary Data 2). IL17A and Th17 CD4 T cells have been reported to mediate allograft rejection[53,54]. However, due to small number of cells in this cluster and individual variability (Supplementary Table 2), it remains to be further investigated whether these cells play a role in tumor rejection.

Clonal expansion is another indicator of tumor reactivity and has been observed in CD4 CTLs in cancer patients[25]. To examine the clonal expansion state of the CD4 subsets, we profiled the TCR repertoire of infiltrating T cells in HT-29 tumors. Out of the 6,712 CD4 T cells, each with one full-length productive α and β chain pair, 5255 clonotypes were detected. C1-PRF1 cells showed a higher frequency of large clones (≥100) and lower Shannon diversity index than Tnaive /Tcm (C5) and Treg subsets (C7, 8 & 9) (Fig. 5G, H), suggesting clonal expansion. Similar observations were made for CD4 T cells in clusters C2-IFIT3 and C4-IL7R.

All CD8 T cell clusters showed various expression levels of cytotoxic molecules (Supplementary Fig. 6A, B), but only C4-CXCL13 was enriched in regressing tumors (Supplementary Fig. 6C). TCR analysis also revealed C4-CXCL13 as the most clonally expanded cluster (Supplementary Fig. 6D). Given that CD8 T cells were dispensable for tumor control, our data suggest that clonal expansion of CD8 T-cell cluster C4-CXCL13 was not sufficient to confer tumor control, potentially due to dysfunction. It is also possible that the increase of this CD8 subset is secondary to an inflamed tumor microenvironment.

**CD4 T cells mount specific HLA-II-restricted killing of HT-29 tumors.** To examine the ability of CD4 T cells to directly kill tumor cells, we co-cultured splenic CD4 T cells from mice that rejected tumors with HT-29 tumor cells ex vivo and evaluated tumor cell survival (Supplementary Fig. 7A). CD8 T cells from the same mouse were used for comparison. To enrich HT-29-reactive T cells, purified CD3 T cells were first expanded with HT-29 tumor cells for 2 weeks prior to the killing assay. Notably,

T cells from mice with tumor regression showed much greater expansion compared to mice with tumor progression, and tumor naive T cells did not expand at all (Supplementary Fig. 7B). The poor expansion of T cells from tumor-progressing mice suggests either their dysfunction or lack of tumor-reactive T cell clones. Both CD4 and CD8 T cells were able to kill HT-29 tumor cells. However, CD4 T cells demonstrated more robust cytotoxicity at lower effector:target ratios than corresponding CD8 T cells (Fig. 6A). Neither CD4 nor CD8 T cells killed RKO, an HLA-I mismatched human colorectal tumor cell line, or Raji, an HLA-II+ mismatched non-Hodgkin lymphoma cell line, confirming that the killing was HT-29 specific. The killing assay was also performed with T cells from regressing tumors (Supplementary Fig. 7C). Similar to splenic T cells, tumor-derived CD4 T cells were enriched by HT-29 co-culture, whereas tumor CD8 T cells did not expand (Supplementary Fig. 7D), again suggesting either their dysfunction or lack of tumor-reactive T cell clones. Consistent with results from splenic T cells, CD4 T cells from regressing tumors showed HT-29 tumor-specific killing. Importantly, the killing was completely abolished by blockade of HLA-II, but not HLA-I, demonstrating that CD4 CTL-mediated tumor killing requires HLA-II recognition (Fig. 6B).

To explore the molecular mechanism of CD4-mediated killing, we examined the expression of cytotoxic molecules on CD4 T cells after HT-29 co-culture and tested the functional requirement of cytotoxic activity in killing the tumor cells. HT-29 tumor cell killing by tumor CD4 T cells was accompanied by markedly increased intracellular expression of perforin and GZMB, as well as the surface expression of CD107a, a marker for degranulating CTLs (Fig. 6C, D). The majority of the CD4 T cells in the HT-29 tumor cell killing assay were PD1+IL7R- CD107a+ (Supplementary Fig. 7E), resembling the phenotype of the intratumoral C1-PRF1 CD4 T-cell population identified by scRNAseq (Fig. 5D, E). Moreover, treating CD4 T cells with concanamycin A (CMA), a well-established inhibitor of the perforin-dependent killing pathway[26], resulted in reduced cytotoxicity of HT-29 tumor cells in vitro (Fig. 6E). These data suggest that the cytotoxic function is required for HT-29 tumor cell killing by the CD4 CTLs. HT-29 co-culture also increased the levels of human proinflammatory cytokines IFNγ, TNFα, and IL-2 in CD4 T cells (Fig. 6F). Consistent with a previous report of TNFα-induced HT-29 cell death[55], a TNFα-neutralizing antibody partially blocked CD4-mediated killing of HT-29 cells (Fig. 6G). Taken together, our data suggest that CD4 CTLs can mount specific, HLA-II-restricted killing of HT-29 tumor cells via the perforin and TNFα pathways.

**Discussion**

We demonstrated here that human CD4 CTLs can mediate spontaneous and potent CD8 T-cell-independent tumor control, resulting in protective memory responses in a preclinical tumor model. Cellular and molecular profiling demonstrates that tumor control is associated with elevated HLA-II expression on tumor cells and increased CD4 CTLs in the blood and tumor of HIS

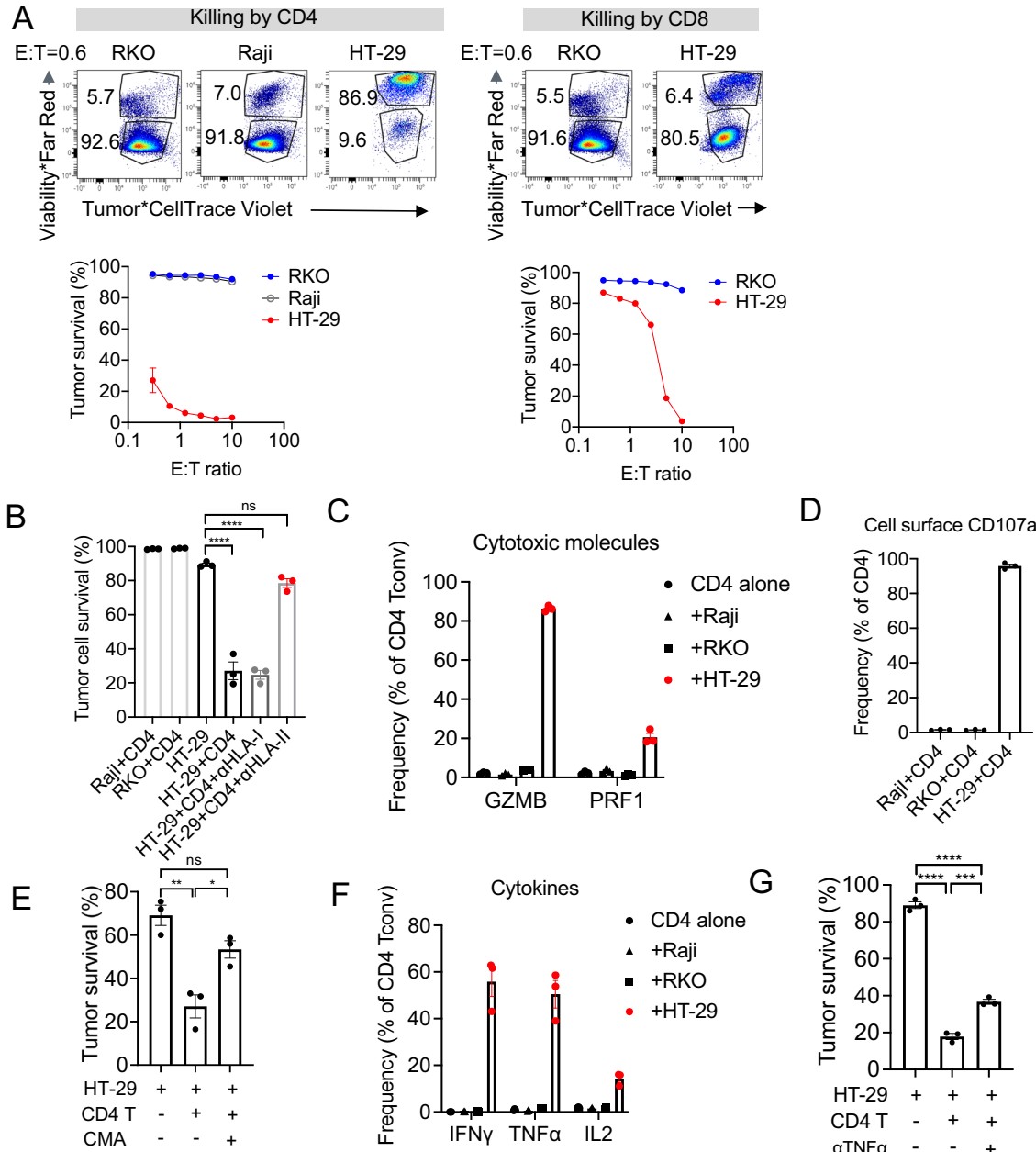

**Fig. 6 CD4 T cells mount specific HLA-II-restricted killing of HT-29 tumor cells. A** Tumor killing assay using splenic CD4 and CD8 T cells of tumor-regressing mice. CD3 T cells were expanded with HT-29 cells for 2 weeks, followed by CD4 and CD8 T cell isolation and tumor co-culture for 3 days. Tumor cell viability was examined as the readout of the assay. RKO and Raji tumor cells were used as negative controls. E:T ratio: effector to target ratio. **B** Tumor killing assay using CD4 T cells from regressing tumors with HLA-II blockade. **C, D** Intracellular staining of cytotoxic molecules at day 2 and cell surface staining of CD107a at the end of the killing assay. **E** Tumor killing assay using CD4 T cells from regressing tumors in the presence of concanamycin A (CMA) to inhibit the perforin pathway. **F** Intracellular staining of proinflammatory cytokines at day 2 of the killing assay. **G** Tumor killing assay using CD4 T cells from regressing tumors in the presence of anti-TNFα. N = 3. The results are shown as means ± SEM. *p < 0.05, **p < 0.01 ***p < 0.001, ****p < 0.0001, one-way ANOVA, Tukey's test.

mice. The intratumoral PRF1[+] CD4 CTL subset enriched in regressing tumors exhibits a strong effector phenotype and clonal expansion. CD4 T cells are necessary for tumor control in vivo, as shown by the in vivo T-cell depletion experiments. Further, these cells are also necessary and sufficient to kill HT-29 tumor cells ex vivo, although we cannot rule out contributions from other cell types in vivo. Based on the ex vivo mechanistic studies, CD4 T-cell-mediated cytotoxicity is dependent on the perforin and TNFα pathways. Taken together, our study provides compelling evidence for CD8-independent tumor control by human CD4

T cells in vivo and offers a valuable preclinical tool to explore therapeutic strategies to enhance CD4 CTL differentiation and function.

CD4 CTL subsets identified in this study closely resemble the phenotype of CD4 CTLs detected in cancer patients. In both cases, the cells express a core gene set of cytotoxic molecules (GZMA, GZMB, PRF1, GNLY, and NKG7), chemokines (CCL4 and CCL5), and Th1 markers (IFNG and CXCL13)[24–26,56,57]. CXCL13[+] CD4 T cells from both HIS mice and patient tumor samples consist of heterogeneous populations, including CTLs

that express cytotoxic, coinhibitory, and Th1 markers[57]. The CD4 CTL clusters in our studies called FC7-GZMB & GNLY (identified by flow cytometry) and C1-PRF1 (identified by scRNAseq) are likely the same population based on similar expression patterns of cytotoxic molecules, activation/memory/proliferation markers, immune checkpoints, and their enrichment in regressing tumors. One minor difference is that C1-PRF1 expresses high CXCL13 mRNA, whereas FC7 expresses only a modest level of CXCL13 protein. The discrepancy is likely due to the secretion of CXCL13 protein in tumor-specific CTLs upon TCR stimulation[58,59].

In addition to CTL phenotypes, FC7-GZMB & GNLY and its scRNAseq counterpart C1-PRF1 exhibited other similarities to the anti-tumor T-cell populations reported in human tumor studies. These cells express PD1 and ICOS, similar to the ICOS$^+$PD1$^+$ Th1-like population that was expanded upon CTLA-4 blockade in melanoma patients[60]. Their expression of PD1[46] and CXCL13[57] is reminiscent of tumor-specific T cells in patients. In treatment-naive patient tumors, tumor-specific T cells show proliferation and exhaustion phenotypes, presumably due to constant antigen stimulation[46]. FC5-Tex cells, which by our flow cytometry analysis were found to upregulate Ki67, may contain some exhausted tumor-specific T cells that fail to clear tumors. In comparison, if cells in the FC7-GZMB & GNLY and C1-PRF1 clusters represent functional tumor-specific T cells that have effectively cleared tumors, they would no longer experience antigen stimulation and thus do not express Ki67.

In addition to phenotypic similarities, CD4 CTL subsets identified in our study also exhibit functional similarities to the anti-tumor T-cell populations detected in cancer patients. In terms of tumor control mechanisms, both HIS- and patient-derived CD4 CTLs exhibit specific HLA-II dependent tumor cell killing ex vivo, accompanied by the release of cytotoxic granules and proinflammatory cytokines[25,26]. Human CD4 T cells might also contribute to tumor control by inhibiting tumor cell proliferation[61] and up-regulating CXCL13 expression, which in turn promotes TLS formation and T cell priming[24,48,52]. Furthermore, it has been reported in mice that CD4 T cells may indirectly kill cancer cells via tissue remodeling and hypoxia-associated mechanisms[62,63].

HIS- and patient-derived CD4 CTLs both associated with tumor growth control: in melanoma patients, a CD4 cluster co-expressing CXCL13 and cytotoxic genes was associated with better overall survival;[57] in colorectal cancer patients, a GZMB$^+$ Th1-like CD4 subset was enriched in PD1-responding MSI tumors;[24] in bladder cancer patients, a gene signature of cytotoxic CD4 T cells in tumors predicted clinical response to anti-PD-L1 treatment;[56] and in our model, increased CD4 CTLs in blood and tumors associated with potent tumor control, including complete tumor rejection. Prior to our study, an outstanding question of recent clinical findings was whether CD4 CTLs are important to tumor control in patients. By depleting CD4 and CD8 T cells separately, our study showed that human CD4 T cells mediated potent tumor control independent of CD8 T cells, implying that CD4 CTLs can play a critical and non-redundant role in fighting particular HLA-II$^+$ tumors.

One intriguing observation of our study was that CD4-mediated tumor control resulted in potent memory responses that protected mice from tumor rechallenge. Like the primary response, the memory was presumably also mediated by CD4 CTLs; however, the role of CD8 CTLs has yet to be ruled out. Primary HT-29 tumor rejection resulted in robust eradication of HCT116 tumors with mismatched HLA-II except for DPA1. This result supports the possibility that CD4 T cells reject HT-29 tumors through DPA1 allo-recognition. Alloantigen-specific T cells have been shown to mediate potent tumor control[64,65]. Our data suggest that CD4 CTLs might be induced by tumor vaccination and that alloantigen-specific CD4 CTLs can potentially be harnessed for the development of T-cell therapies. Exploring these possibilities may provide new treatment options for HLA-II$^+$ cancers.

Another intriguing observation of our study is that CD4 T cells are required whereas CD8 T cells are dispensable for HT-29 tumor control. CD8 T cells are critical for the control of multiple tumor types. Nevertheless, a role for CD4 CTL in anti-tumor immunity is increasingly being appreciated, especially in HLA class II$^+$ tumor types in patients. Our study provides a model with human T cells and human tumor cells to further investigate the anti-tumor activity of CD4 CTLs. In our model, the difference in CD4 versus CD8 T cell activity is unlikely due to cell number, as CD4 to CD8 T cell ratio was not associated with tumor outcome. A potential explanation for the lack of CD8 T cell activity is the requirement for additional co-stimulatory signals to trigger CD8 CTL killing against HT-29, a HLA-II$^+$ colorectal cell line lacking expression of common co-stimulatory molecules such as CD80/CD86. In support of this hypothesis, CD8 T cells in HIS mice have been shown required to control the growth of Raji, a B lymphoma cell line expressing high levels of co-stimulatory molecules (ref. [66] in press). These findings suggest that human immune cells exploit distinct mechanisms to target different tumors, and that CD4 abundance in the HIS model does not preclude CD8 T cell in clearing tumors. These studies support the therapeutic strategy to enhance anti-tumor immunity by boosting CD4 CTLs, especially when CD8 T cells become dysfunctional, or tumor cells lack HLA-I/co-stimulatory molecules.

Our work shows that CD4 CTLs can potently kill tumors and can independently mediate tumor regression, but there are limitations. For instance, although a variety of tumor types have been shown to be HLA-II positive, HLA-II expression on tumor cells is less common and homogeneous than HLA-I expression[9]. In addition, tumor cells can likely also escape MHC-II-mediated killing through genetic mutations, epigenetic silencing, or post-translational modifications downregulating the HLA-II antigen-processing and presentation machinery[67,68]. Understanding the regulation of HLA-II expression by tumors is critical for developing CD4 CTL-targeting therapeutic strategies. Chronic antigen exposure and immune suppression by Tregs or myeloid-derived suppressive cells render CD8 CTL dysfunctional[69–71]. Similarly, understanding the mechanisms of CD4 CTL dysfunction is critical to enhancing their anti-tumor activity.

In summary, we characterize a model in which human CD4 CTLs play an independent, essential, and non-redundant role in tumor control. This work provides a strong rationale to enhance tumor killing by boosting CD4 CTLs, especially in the setting of CD8 T-cell dysfunction and/or MHC-I downregulation.

## Methods

**HIS mice**. The generation of knock-in mice encoding human SIRPA and TPO in a 129xBALB/c genetic background was performed using the Velocigene technology®[34,35,72]. The mice were crossed to a Rag2$^{-/-}$ Il2rg$^{-/-}$ background to generate the SIRPA$^{h/h}$ TPO$^{h/m}$ Rag2$^{-/-}$ Il2rg$^{-/-}$ (StRG) mice that are homozygous for human SIPRA and heterozygous for human TPO. HIS mice were generated by engrafting irradiated newborn or 4-week-old StRG mice with $5 \times 10^4$ to $1 \times 10^5$ human CD34$^+$ HSPCs isolated from fetal liver or cord blood, respectively. Fetal liver CD34$^+$ HSPC were obtained from Advanced Biosciences Resources (Alameda, CA) with proper consent[66] and cord blood CD34$^+$ HSPC were obtained from AllCells, HemaCare, or STEMCELL Technologies. Human immune cell reconstitution was confirmed using flow cytometry 16–24 weeks after HSPC engraftment. The mice were kept on sulfatrim diet to prevent bacterial infection and maintained under pathogen-free conditions. All experiments involving mice were performed in compliance with all relevant ethical regulations and following protocols approved by the Regeneron Pharmaceuticals Institutional Animal Care and Use Committee (IACUC).

**Tumor studies**. The base medium for HT-29 and HCT116 was McCoy's 5a Medium Modified and that for CA46 was RPMI-1640. All base media were supplemented with 10%

FBS (Gibco #26170043) and 1 mM penicillin–streptomycin (Gibco #15140148). Tumor cells were cultured in a humidified $CO_2$ incubator at 37 °C, maintained at ~70% confluency, harvested with 0.25% (w/v) Trypsin-0.53 mM EDTA solution, and suspended in DPBS with a concentration of $2 \times 10^7$ cells/ml for injection. HIS mice and non-reconstituted StRG mice were injected with $2 \times 10^6$ tumor cells subcutaneously (s.c.) into the right flank. Tumor volumes were measured using a digital caliper and calculated using the formula $L \times W \times W \times 0.5$ where $L$ was the longest dimension and $W$ was the perpendicular dimension. Tumor regression was defined as having two tumor volume values smaller than the preceding peak value.

To deplete T cells, HIS mice were treated with mIgG2a (Clone C1.18.4, BioXCell #BE0085) plus mIgG2b (Clone MPC-11, BioXCell #BE0086), anti-CD4 (OKT-4, BioXCell #BE0003-2), anti-CD8α (Clone OKT-8, BioXCell #BE0004-2), or anti-CD4 plus anti-CD8α. Each mouse was injected with 250 μg of the antibodies two to three times a week, The treatment started 9 days before tumor implantation and lasted throughout the tumor study.

**Flow cytometry.** For sample preparation, 200 μl blood was collected into 500 μl of 2% dextran. The supernatant containing peripheral blood mononuclear cells were collected for downstream analysis. Spleens were dissected from mice and kept in DPBS on ice. Single-cell suspensions were prepared by mechanical dispersion, and red blood cells were lysed using Gibco™ ACK Lysing Buffer (Thermo Fisher Scientific #A1049201). Tumors were dissected from mice, cut into small pieces of 2–4 mm, transferred into a dissociation buffer (Mouse Tumor Dissociation Kit, Miltenyi #130-096-730), and kept on ice. Single-cell suspensions were prepared by enzymatic digestion and mechanical dispersion using gentleMACS™ Octo Dissociator with Heaters (Miltenyi #130-096-427). One million splenocytes or tumor cells were used for flow cytometry analysis.

For staining, LIVE/DEAD™ Fixable Blue (Thermo Fisher Scientific #L34962) or Zombie UV™ Fixable Viability dye (BioLegend #423108) was used to discriminate live and dead cells. Human TruStain FcX™ (BioLegend #422302) and TruStain FcX™ PLUS (anti-mouse CD16/32 Antibody (BioLegend #156604) were used to block human and mouse Fc receptors. Staining Buffer (BioLegend #420201) and Brilliant Stain Buffer Plus (BD #566385) were used to diluting antibodies for cell surface marker staining. Intracellular marker staining was performed using the eBioscience™ Foxp3/Transcription Factor Staining Buffer Set (Thermo Fisher Scientific #00-5523-00). Antibodies for staining are listed in Supplementary Table 3.

Flow cytometry analysis was performed using Cytek® Aurora (5 Laser), and data analysis was performed using FlowJo™ v10.7.1 and OMIQ.

**scRNAseq and TCRseq sample preparation.** One pair of tumor-progressing and tumor-regressing mice were processed each day. Tumors were dissociated into single-cell suspensions as described above and incubated with human CD326 (EpCAM) MicroBeads (Miltenyi #130-061-101) in the presence of human and mouse Fc blocker. The samples were then passed through the autoMACS® Pro Separator (Miltenyi #130-092-545) to deplete tumor cells. The negative fraction was stained with anti-hCD45 and sorted for hCD45-positive cells using BD FACSymphony™ S6.

**5′ single-cell partitioning with GEX and TCR library preparation, sequencing, and read alignment.** Single cells suspended in PBS with 0.04% BSA were loaded, 10,000 cells per lane, on a Chromium Connect Single-Cell Liquid Handler (10x Genomics). RNA-seq and VDJ libraries were prepared using Chromium Next GEM Automated Single-Cell 5′ Kit, v2 (10x Genomics). After amplification, cDNA was split into separate RNA-seq and VDJ aliquots. To enrich the VDJ aliquot for TCR sequences we used the 10x Genomics Chromium Automated Single-Cell Human TCR Amplification & Library Construction Kit (10x Genomics). Paired-end sequencing was performed on Illumina NovaSeq 6000 for RNA-seq libraries (Read 128 bp for UMI and cell barcode, Read 280-bp for transcript read, with 10-bp i7 and 10-bp i5 reads) and for VDJ libraries (Read 1 150-bp, 10-bp i7, 10-bp i5, Read 2 150-bp). For RNA-seq libraries, Cell Ranger Single-Cell Software Suite (10x Genomics, v2.2.0) was used to perform sample demultiplexing, alignment, filtering, and UMI counting. The human GRCh38 and mouse mm10 genome assembly and RefSeq gene model for human and mouse were used for the alignment. For VDJ libraries, Cell Ranger Single-Cell Software Suite (10x Genomics, v2.2.0) was used to perform sample demultiplexing, de novo assembly of read pairs into contigs, align and annotate contigs against all of the germline segment VDJ reference sequences from human IMGT, label and locate CDR3 regions, group clonotypes.

**scRNAseq data analysis (dimensionality reduction, unsupervised clustering, cluster annotation, and comparison of cluster proportion).** The analysis was carried out using version 3 of the Seurat R package[73]. Cells with >500 genes mapped to the mouse genome, <900 genes mapped to the human genome, or >20% reads mapped to mitochondrial genes were discarded from the analysis. Gene expression values for each cell were normalized and scaled due to variation in the cell-cycle stage. The number of UMI was also regressed out to correct for variation in sampling depth of these cells. The genes used for principal component analysis were the 2000 genes with the highest variance, mean UMI between 0.0125 and 8, and dispersion above 0.5. Genes were divided into seventeen bins of equal width

based on their average expression and dispersion. Z scores were calculated within these bins. Cells were then partitioned into clusters (Seurat FindClusters function) and visualized using the Uniform Manifold Approximation and Projection (UMAP) algorithm (Seurat RunUMAP function). The first fifteen principal components were used to run the UMAP dimensionality reduction. The FindClusters function was run with a resolution parameter of 0.7 for hCD45 clustering, resulting in eleven clusters of cells. Cluster cell type identities were annotated by differentially upregulated genes of each cluster compared with the rest of the cells (Seurat FindAllMarkers function; significance is defined as $p < 0.01$ and fold change >1.5) and expression of canonical immune marker genes (Seurat FeaturePlot function). The T cells and prolif T & NK cells with CD4 and CD8 UMI > 1 were stratified and re-clustered. Using a resolution parameter of 0.6, six clusters were identified, including CD4, CD8, prolif T-MCM, prolif T-MKI67, Treg, and prolif Tregs. The CD4 cluster and cells in the two proliferating (prolif) clusters with CD4 UMI > 1 were further subsetted and re-clustered to identify specific CD4 sub-cluster populations. The CD8 cluster and cells in the prolif clusters with CD8 UMI > 1 were further subsetted and re-clustered to identify specific CD8 sub-cluster populations.

Comparison of cluster proportion between tumor-progressing and tumor-regressing mice was performed using the scProportionTest function[74]. This function uses a bootstrapping permutation test to calculate a $p$ value, fold change and confidence interval for each cluster to identify the magnitude of difference between samples. The difference was considered statistically significant if the adjusted $p$ value (FDR) was <0.01 and the absolute value of the fold change was >1.5.

**scTCRseq data analysis.** Clone size was calculated as the number of TCRs per sample. We summarized the number of clones into bins of different clone sizes for visualization. Diversity measures based on clonotypes were presented as Shannon diversity index, calculated using the clonalDiversity function of scRepertoire[75].

**Ex vivo functional assays.** Splenic T cells were purified using human CD3 MicroBeads (Miltenyi #130-050-101) and co-cultured with irradiated HT-29 cells at a 4:1 ratio. 5 ng ml$^{-1}$ of IL7 and IL15 were added at D0, and 10 U ml$^{-1}$ IL-2 was added at D2. Cytokines were replenished every 2 days until day 14 when the cells were frozen down. Live HT-29 cells were added at a splenocyte-to-tumor ratio of 4:1 on day 7.

The day before the tumor-killing assay, tumor cells were labeled with 2.5 μM CellTrace Violet dye (Thermo Fisher Scientific #C34557) at room temperature for 20 minutes, resuspended in T Cell Medium (TCM; RPMI media supplemented with beta-mercapthoethanol, HEPES buffer, non-essential amino acids, sodium pyruvate, 10% FBS, and 1 mM penicillin-streptomycin) with a concentration of $7 \times 10^5$ cells ml$^{-1}$, and plated in flat-bottom 96-well plates with 100 μl per well. On the day of the assay, media of the tumor cells was changed to remove dead cells. T cells were thawed, and CD4 and CD8 T cells were separated using human CD8 MicroBeads (Miltenyi #130-045-201). 100 μl CD4 or CD8 T cells in TCM were added into tumor cells at a series of E:T ratios (10:1, 5:1, 2.5:1, 1.25:1, 0.625:1, and 0.3125:1). 1 μg ml$^{-1}$ anti-CD28 (BioLegend #302934) was then added, and the cells were cultured in a humidity-controlled incubator for 3 days. To block HLA, anti-HLA class I (10 μg ml$^{-1}$, clone W6/32, BioLegend 311428) or anti-HLA class II (5 μg ml$^{-1}$ each of clone L243, BioLegend #307648 and clone Tü39, BioLegend #361702) was added 2 hours before the co-culture. To inhibit the perforin pathway, CD4 T cells were treated with 100 nM of concanamycin A (TOCRIS #2656) for 2 hours, washed with TCM, and co-cultured with tumor cells. To block the TNFα pathway, 2 μg ml$^{-1}$ of a TNFα neutralizing antibody (Cell Signaling 7321 S) was added 2 hours before the co-culture. 3 days after the co-culture, both floating and attached cells were collected, and the cells were stained with LIVE/DEAD™ Fixable Far Red Dead Cell Stain (Thermo Fisher Scientific #L34974) to evaluate tumor cell death and survival.

To perform tumor killing assay using T cells from tumors, CD4 and CD8 T cells were sorted as CD2$^+$CD4$^+$ and CD2$^+$CD8$^+$ fractions, respectively. Each fraction was expanded with HT-29 for 21 days and the assay was performed as described above.

To examine the production of effector molecules, protein transport inhibitors (BD #554724 and #555029) were added 2 days after the co-culture, and cells were collected 5 hours afterwards for CD3, CD4, and CD8 cell surface staining, followed by intracellular staining of FOXP3 and effector molecules. To evaluate degranulation and release of cytotoxic molecules, CD107a cell surface staining was performed at the end of the killing assay.

**Statistics and reproducibility.** Significance values ($p$ values) were calculated with unpaired two-tailed Student's $t$ test for two-group comparisons or one-way ANOVA with Tukey's test for multigroup comparisons. In cases of two variables, two-way ANOVA with Sidak's multiple comparison was used for comparisons under each variable. Correlation between two variables was evaluated using the Pearson correlation coefficient. The effect of sex on tumor growth outcome was evaluated using Chi-square test. $p$ values of <0.05 were considered significant. *$p < 0.05$, **$p < 0.01$, ***$p < 0.001$, ****$p < 0.0001$. Statistical analyses were performed using GraphPad Prism™ v8. Sample sizes are indicated in each figure

legend. Measurements were taken from distinct samples unless otherwise indicated in the figure legends. Tumor rejection phenotypes were repeatedly observed in 27 independent cohorts engrafted with different CD34[+] HSPC donors. In vitro killing was repeated three times with T cells isolated from different HIS mice.

**Reporting summary**. Further information on research design is available in the Nature Portfolio Reporting Summary linked to this article.

## Data availability

The source data underlying all graphs and charts are provided in Supplementary Data 3. The single-cell RNA sequencing and TCR sequencing data have been deposited in NCBI's Gene Expression Omnibus (Edgar et al., 2002) and are accessible through GEO Series accession number GSE223026 (https://www.ncbi.nlm.nih.gov/geo/query/acc.cgi?acc=GSE223026). Any remaining information can be obtained from the corresponding author upon reasonable request.

## Code availability

The code used in this study can be accessed through the following link: https://github.com/gchoonoo/WenLin_scs_2023.

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

## Acknowledgements

We thank Funmi Adewale, Sean D'Italia, and Chan-Jung Chang for their assistance with cell sorting; Benjamin Daniel, Sean D'Italia, and Neil Patel for their advice on flow cytometry panel design; Lance Zhang for performing scRNAseq; Yi Wei, Min Ni, and Christian Adler for their advice on scRNAseq data analysis; Melanie Buckman and Michelle McAlister for colony management of immune-deficient mice; Lauric Haber, Arpita Pawashe, Stephane Pourpe, and Se Jeong for their advice on ex vivo assays; members of the Regeneron Immuno-Oncology group and the postdoctoral committee for their input throughout the project.

## Author contributions

W.L. and Y.L. designed the study. W.L., V.S., and R.S. performed the experiments. W.L., G.C., Y.L., and N.G. analyzed the data. A.P. and D.F. provided HIS mice. J.Z. provided immune-deficient mice. L.M. supported HIS mice generation and molecular profiling work. C.D., T.O., and A.M. provided insight into the project. E.I., M.M., and G.T. supervised the study and provided advice. W.L. and Y.L. wrote the manuscript.

## Competing interests

This study was sponsored by Regeneron Pharmaceuticals, Inc. All authors are or used to be employees of Regeneron and may hold stock options in the company.
