## [Peer Review File · Communications Biology]

This manuscript has been previously reviewed at another Nature journal.

REVIEWERS' COMMENTS:

Original Reviewer #3 (Remarks to the Author):

Manuscript Nr: COMMSBIO-23-0717-T

Lin et al., "Human CD4 cytotoxic T lymphocytes mediate potent tumor control via tumor cell HLA class II expression in a humanized immune system mouse model"

The authors demonstrate that the subcutaneous transplantation of the human colorectal cancer cell line HT-29 into mice with reconstituted human immune systems (humanized mice), leads to tumor regression in nearly half of the transplanted mice. Rechallenge of the HT-29 with the same tumor leads to higher rejection levels and the protection lasts at least 80 days. Rejection was found to be dependent on CD4+ but not CD8+ T cells and the authors observe CD4+ T cells that kill HT-29 cells ex vivo. They then compare by single cell RNA sequencing, regressing with non-regressing tumors and observe a correlation between T cell infiltration (non-proliferating CD8+ T cells and 9 subpopulations of CD4+ T cells) with tumor regression. 3 regulatory T cell populations were preferentially found in progressing tumors. 3 cytotoxic CD4+ T cell populations were observed in regressing tumors. In vitro the authors determine that CD4+ T cells from HT-29 rejecting mice kill HT-29 but not other allogeneic tumor cells that are HLA-mismatched with HT-29. The authors claim that their study is the first to demonstrate protective effects of cytotoxic CD4+ T cells.

In their revised manuscript version, the authors have addressed all of my concerns. They employ rechallenge with two additional tumor cell lines, one that shared HLA-DPA1 as the only MHC class II alloantigen and could, therefore, mediate both HT-29 and HCT116 rejection. Furthermore, they provide degranulation assays to demonstrate that PD1+IL7R- CD4+ T cells primarily released their cytotoxic machinery in response to HT-29 stimulation, and could represent the C1-PRF1 population. Finally, the authors could block some tumor cell killing with concanamycin A, blocking acidification of cytotoxic granules and thereby their reactivity. In addition, the authors have toned down their claim to be able to identify new tumor antigens with the presented allogeneic tumor cells challenge. With these significant revisions I find the manuscript sufficiently improved.

Comment on authors' rebuttal to original Reviewer #2:

I had also a look at their response to the other reviewer. In my opinion the authors make a good case that not all highly PD-1+ CD4+ T cell can be lumped together and that only the GZMB+ PD-1+ CD4+ T cell population correlated with tumor regression in their model. They also clarify the discrepancy between CXCR13 mRNA and protein in this FC7 cluster, as CXCL1 protein could have been released due to antigen encounter, thereby lowering the flow cytometric staining in a CXCL13 mRNA high population. They also address the peculiarity that their HT-29 tumor is particularly sensitive to CD4+ and not CD8+ CTL by referring to their other study in press, in which RAJI tumors are primarily rejected by CD8+ T cells. This indicates that HIS mice have not only functional CD4+ T cells. They also relate their CXCL13+ CD4+ T cells to another study in melanoma and provide some discussion why upon CD8+ T cell depletion immune control of HT-29 even slightly increases, even so this effect is not very pronounced to start with. Therefore, I find that the authors also sufficiently addressed the concerns of the other reviewer.

Overall I would recommend acceptance of this study.

REVIEWER COMMENTS

Reviewer #2:

In this manuscript, the authors used several experimental systems to investigate CD4 T cell-mediated anti-tumor immunity. Using the humanized immune system mouse model, the authors found that tumor-infiltrating CD4 effector T cells were critical in the control of tumor and the depletion of CD4 T cells rather than CD8s significantly promoted tumor progression. Additionally, ex vivo assays further revealed that CD4 T cells could directly mediate target killing of cancer cells in an HLA-II-dependent manner. While the work is interesting, I have several concerns relating to this study, and in particular the phenotypes and characteristics of those cytotoxic CD4 T cell subsets.

Major points:

1. One major concern with the study is the unclear description and characterization of cytotoxic CD4 T cells. In Fig. 4 and S3, the authors identified FC6-CXCL13, FC5-Tex and FC7-GZMB+GNLY+ subsets, and found that it was FC7-GZMB+GNLY+ cells rather than FC5-Tex cells that were enriched in regressing tumors. The authors tried to label FC7-GZMB+GNLY+ cells as a cluster distinct from FC5 and FC6. However, based on Fig.S3D, these three subsets (FC5, FC6 & FC7) were highly similar in terms of the high expression of PD1, TIM3, CTLA4 and LAG3.

In addition, it is unclear to me whether the FC7-GZMB+GNLY+ cells characterized by flow cytometry and the C1-PRF1 & C3-KLRB1 cells identified by scRNA-seq (Fig. 5) were the same population. C1-PRF1 & C3-KLRB1 cells detected by scRNA-seq were found to be enriched in regressing tumors and, intuitively, such cells correspond to those FC7-GZMB+GNLY+ cells. However, C1-PRF1 & C3-KLRB1 cells highly expressed CXCL13 while FC7-GZMB+GNLY+ cells did not. Accordingly, the authors need to address this controversy. My suggestion is to treat FC5, 6 & 7 in Fig. 4 as one cell population because all these cells highly expressed PD1 (Fig. 4B), which is likely due to persistent exposure to its cognate tumor antigen in the tumor microenvironment. If this combined cell population increased in regressing tumors, the authors could then map this cell population to C1-PRF1 & C3-KLRB1 cells (also highly expressed PDCD1, Fig. 5E), and this increase would be consistent with the expansion of ICOS+PD1+ Th1-like cells upon CTLA-4 blockade (Wet S et al., Cell 2017). The authors need to note that the upregulation of dysfunctional signatures is only one of the phenotypes of tumor-specific T cells, and those T cells (at least for CD8s) previously defined as exhausted are in fact a highly proliferating T cell population and are likely to be the major T cell population to mediate the anti-tumor immunity within treatment-naive tumors (Li H et al., Cell 2019, Caushi et al., Nature 2021, Thommen et al., Nat. Med. 2018 and Duhon et al., Nat Commun 2018).

We thank the reviewer for the insightful comments and suggestions. We examined FC5, 6, & 7 as a combined population and did not observe any increase in regressing tumors, possibly because these clusters represent cells of different functional states despite their common expression of PD1 and other immune checkpoints.

- Cells in cluster FC5 express high levels of Ki67, which has been associated with exhausted T cells with impaired anti-tumor activity (Miller, Sen et al. 2019). Consistently, this population is enriched in progressing tumors (Supplementary Fig. 4E), suggesting that FC5 represents the exhausted T cells that fail to clear tumors. In treatment naïve patient tumors, tumor-specific T cells show proliferation and exhaustion phenotypes, reportedly due to constant antigen stimulation (Thommen, Koelzer et al. 2018). Accordingly, FC5 may contain some exhausted tumor-specific T cells.
- Cells in cluster FC6 highly express CXCL13, a chemokine that mediates immune cell recruitment to tertiary lymphoid structure (Petitprez, de Reynies et al. 2020). In patient tumors, CXCL13 protein is expressed by Th1, Tfh, and exhausted CD8 T cells (Gu-Trantien, Loi et al. 2013, Thommen, Koelzer et al. 2018, Cohen, Giladi et al. 2022). Therefore, FC6 may represent cells with Th1 or Tfh function. The frequency of this cluster was similar between progressing and regressing tumors.
- Cells in cluster FC7 express well-established cytotoxic markers GZMB and GNLY and are enriched in regressing tumors (Fig. 4D). This CD4 cluster may represent the CD4 CTLs that mediate tumor control via direct cytotoxicity (Oh, Kwek et al. 2020, Cachot, Bilous et al. 2021).

Descriptions of these three populations are now added in lines 203-215 and 369-375.

Cluster FC7 is likely the equivalent of C1-PRF1 identified by scRNAseq, based on similar expression patterns of cytotoxic molecules (GNLY, GZMB), activation/memory/proliferation markers (CCR7, IL7R, HLA-DR, Ki67), immune checkpoints (PD1, CTLA4, TIM3, LAG3) (Fig. 4B, 5D, Supplementary Fig. 4C, Supplementary Fig. 5D & Additional Fig. 1), and their enrichment in regressing tumors (lines 355-361). As pointed out by the reviewer, C1-PRF1 expresses high CXCL13 mRNA whereas FC7 expresses only a modest level of CXCL13 protein (Additional Fig. 2). The discrepancy is likely due to secretion of CXCL13 protein in tumor-specific CTLs upon TCR stimulation (Workel, Lubbers et al. 2019, Li, Wu et al. 2021) (lines 361-363).

In addition to CTL phenotypes, cluster FC7 and its scRNAseq counterpart C1-PRF1 exhibited other similarities to the anti-tumor T cell populations reported in human tumor studies. These cells express PD1 and ICOS (Additional Fig. 3), similar to the ICOS⁺PD1⁺ Th1-like population expanded upon CTLA-4 blockade in melanoma patients (Wei, Levine et al. 2017). Their expression of PD1 (Thommen, Koelzer et al. 2018) and CXCL13 (Veatch, Lee et al. 2022) is reminiscent of tumor-specific T cell in patients. However, unlike the exhausted tumor-specific T cells in treatment naïve patients and in cluster FC5, cells in cluster FC7 likely represent functional tumor-specific T cells that effectively

cleared tumors. Therefore, they are not under constant antigen stimulation and do not express Ki67 (lines 365-375).

Together, these further characteristics of CD4 CTLs are now discussed in lines 355-375 and new Additional Figures 1-3.

Additional figure 1. UMAPs of additional signature gene expressions in tumor-infiltrating CD4 T cell clusters. C1-PRF1 cluster is CCR7⁻, IL7R⁻, and HLA-DRB1⁺.

Additional figure 2. Flow cytometric analysis of GNLY and CXCL13 expressions in different tumor-infiltrating CD4 T cell clusters. Each plot is a concatenation of all tumor-bearing mice. FC8-Treg is used as negative gating control for GNLY and CXCL13. Majority of CXCL13⁺ cells in FC7-GNLY & GZMB co-express GNLY and represent ~20% of the cluster.

Additional figure 3. UMAPs of PD1 and ICOS expressions in tumor-infiltrating CD4 T cells. Conventional CD4 T cells, including the PRF1 subset, express PD1 and ICOS.

2. In Fig. 2, regressing tumors had higher abundance of CD8s and lower level of Tregs compared to progressing tumors, and the differences were notable. It is unclear to me why the difference was smaller when calculating the CD8/Treg ratio.

Since Treg counts are similar (Additional Fig. 4), the difference in CD8/Treg ratio is mostly driven by CD8 counts. The significance is reduced due to two statistically defined outliers; one belongs to the progressor group and another to the regressor group (Grubbs' test, $p < 0.05$). Excluding these two outliers makes the difference in CD8/Treg ratio between the P and the R groups more prominent but does not change the overall message; therefore, we report all 186 data points in Fig. 2C.

Additional figure 4. Comparison of Treg count between tumor-progressing (P) and tumor-regressing and rejected (R) mice prior to tumor implantation. Treg count was similar between the two groups.

3. It would be helpful to show proportions of immune cell types for all the n=21 HIS mice samples in Fig. S1B (e.g., boxplot). In addition, the authors need to show the frequency of CD4 and CD8 T cells where possible. In Fig. S2C (the middle panel), 75% T cells were CD4 cells (isotype), and this seems to be abnormal and could possibly explain the central role of CD4 T cells in the mouse model shown in this manuscript.

We thank the reviewer for these suggestions. The bar graph is now shown as scatter plot in the new Supplementary Fig.1A and as box plot (Additional Fig. 5). The CD4 and CD8 frequencies as well as their ratio for the 184 mice of Fig. 2A (including the 21 mice of Supplementary Fig. 1A) are now shown in the new Supplementary Fig. 2B.

**Immune composition in the blood
(HIS mice)**

Additional figure 5. Proportions of immune cell types in the blood of HIS mice (n=21) shown as box plot.

The CD4 to CD8 ratios in our HIS cohorts are usually greater than one, as has been reported in humans (mean 2.13 ± 1.04 and median 1.88 in a study of 468 healthy individuals) (Amadori, Zamarchi et al. 1995). In our studies, the ratio was not associated with tumor outcome (Supplementary Fig. 2B). In addition, CD8 T cells in HIS mice have been shown required in controlling the growth of another tumor type, Raji, a B lymphoma cell line expressing high levels of HLA-II and co-stimulatory molecules (Patel et al. 2023, in press). These findings suggest that human immune cells exploit distinct mechanisms to target different tumors, and that CD4 abundance does not preclude CD8 T cell from clearing tumors in the HIS model.

These concepts are now discussed in lines 136-138 and 416-421 and the new Supplementary Fig. 2B.

4. In Fig. 2D, it seems that the depletion of CD8 T cells was associated with favorable tumor control compared to the isotype group, and the authors need to provide the rationale underlying this observation.

Depletion of CD8 T cells indeed resulted in a somewhat higher rate of tumor rejection. It is plausible that CD8 depletion decreases the consumption of T cell-supporting cytokines in HIS mice, leading to CD4 T cell expansion and favorable tumor control, as reported in rhesus macaques where CD8 T cell depletion leads to an IL15-dependent expansion of memory CD4 T cells (Okoye, Park et al. 2009).

The rationale is now discussed in lines 148-152.

5. The authors performed the pseudotime analysis to indicate the high differentiation degree of C1-PRF1 cells. However, in Fig. 4, those C2-IFIT3 cells highly expressed IL7R (a marker of CD4 T cell precursors as described in this manuscript) but instead showed high differentiation degree, which reduced the confidence of their analysis.

To lend confidence to our analysis, we examined the expression of Tnaive/Tcm and T effector markers over pseudotime. There was an up-regulation of Tnaive/Tcm markers IL7R and TCF7 as well as a down-regulation of T effector markers GZMB and PRF1, consistent with literature (Caushi, Zhang et al. 2021). These analyses are now discussed in lines 266-268 and new Supplementary Fig. 5E.

The C2-IFIT3 cluster is marked by the expression of interferon responsive genes (ISG). It shows high levels of IL7R expression, possibly due to the presence of CD4 CTL precursors and central memory T cells in this cluster. This cluster exhibits high degree of differentiation, consistent with previous studies where type I interferon signaling drives the differentiation and exhaustion of CD8 T cells (Stelekati, Shin et al. 2014, Wu, Ji et al. 2016). In agreement with our observation, ISG⁺ CD8 T cells in patient tumors have been reported to contain IL7R⁺ memory T cells and show a differentiated phenotype (Zheng, Qin et al. 2021).

6. In Fig. 5B, it seems that only a small number of C3-KLRB1 cells were detected and the authors need to show the cell frequency (also for other clusters) for each sample to address whether the difference was biologically meaningful or driven by potential technical bias.

The frequencies of each cluster and sample are now provided in the new Supplementary Table IV. The frequencies of the C3-KLRB1 population vary from sample to sample. We agree that the difference in this population may not be biologically meaningful and decided not to highlight their enrichment in regressing tumors.

The description of this population is now modified in lines 277-279, new Supplementary Table IV.

7. Both the C1-PRF1 and C3-KLRB1 subsets highly expressed CXCL13, while C3-KLRB1 showed lower expression of HAVCR2 and higher expression of IL7R. This heterogeneity within the CXCL13-expressing CD4 T cells in the tumor is consistent with the recent Veatch et al. Cancer Cell 2022 paper, and the increase of C1-PRF1 and C3-KLRB1 cells in regressing tumors is also in agreement with the tumor-reactivity of CXCL13-expressing CD4 T cells. Further discussion to connect previous work to this study would strength this paper.

We thank the reviewer for highlighting the connection between CXCL13⁺ CD4 T cells identified in our study and those from recent studies of melanoma patient tumors (Veatch, Lee et al. 2022). CXCL13⁺ CD4 T cells from both our study and the melanoma study consist of heterogeneous populations, including CTLs that express cytotoxic (PRF1),

coinhibitory (PD1, HAVCR2, LAG3), and Th1 (IFNG) markers (lines 355-357, Fig. 5E and the new Supplementary Fig. 5D). In melanoma patients, CXCL13⁺ CD4 T cells associate with better overall survival, and similarly in HIS mice, the C1-PRF1 subset of CXCL13⁺ CD4 T cells associates with tumor regression (lines 387-389).

Connection between our study and the Veatch study is now made in lines 355-357 and 387-389. Additional connection between our study and previous work is now made in lines 367-368.

8. While this study provided insights into CD4 T cell biology, the authors should note important limitations with respect to the critical role of CD4 rather than CD8 T cells in their mouse models, because CD8 T cells are indispensable in the control of in tumor many cases, especially in multiple human cancer types. The authors should at least discuss this.

We concur that CD8 T cells are critical for control of multiple tumor types. Nevertheless, a role for CD4 CTL in anti-tumor immunity is increasingly being appreciated, especially in HLA class II⁺ tumor types in patients. Our study provides a model with human T cells and human tumor cells, in which cytotoxic CD4 T cells are required for tumor control. In our model, CD4 T cells are required whereas CD8 T cells are dispensable for HT-29 tumor control. A potential explanation for the lack of CD8 T cell activity is the requirement for additional co-stimulatory signals to trigger CD8 CTL killing against HT-29, a HLA-II⁺ colorectal cell line lacking expression of common co-stimulatory molecules such as CD80/CD86. In support of this hypothesis, CD8 T cells in HIS mice have been shown required to control the growth of Raji tumors, a B lymphoma cell line expressing high levels of co-stimulatory molecules (Patel et al. 2023, in press). These findings support the therapeutic strategy to enhance anti-tumor immunity by boosting CD4 CTLs, especially when CD8 T cells become dysfunctional, or tumors lack HLA-I/co-stimulatory molecules.

Discussion of this point is now added in lines 410-423.

Minor point:

9. The authors need to show those important signature genes such as PDCD1, CTLA4, TIGIT, LAG3, IL17A and CXCL13 in the UMAP plot, for example, in Figs. 4, 5, S3 and S4.

UMAPs for these genes are now provided in the new Supplementary Fig. 5D and described in line 239.

10. It would be helpful to separately show the results for analyzed sample where possible, for example, in Fig. 5G, H.

The separate results for Fig. 5G & H are now provided in the Source Data file.

Reviewer #3:

Manuscript Nr: NCOMMS-22-13271 Lin et al., “Human CD4 cytotoxic T lymphocytes mediate potent tumor control via tumor cell HLA class II expression in a humanized immune system mouse model”

The authors demonstrate that the subcutaneous transplantation of the human colorectal cancer cell line HT-29 into mice with reconstituted human immune systems (humanized mice), leads to tumor regression in nearly half of the transplanted mice. Rechallenge of the HT-29 with the same tumor leads to higher rejection levels and the protection lasts at least 80 days. Rejection was found to be dependent on CD4+ but not CD8+ T cells and the authors observe CD4+ T cells that kill HT-29 cells *ex vivo*. They then compare by single cell RNA sequencing, regressing with non-regressing tumors and observe a correlation between T cell infiltration (non-proliferating CD8+ T cells and 9 subpopulations of CD4+ T cells) with tumor regression. 3 regulatory T cell populations were preferentially found in progressing tumors. 3 cytotoxic CD4+ T cell populations were observed in regressing tumors. *In vitro* the authors determine that CD4+ T cells from HT-29 rejecting mice kill HT-29 but not other allogeneic tumor cells that are HLA-mismatched with HT-29. The authors claim that their study is the first to demonstrate protective effects of cytotoxic CD4+ T cells.

Although the reported findings are interesting the described model most likely is still based on alloreactive HT-29 tumor cell rejection and has not proven that cytotoxic functions of the expanded CD4+ T cell stimulations are required for the observed tumor cell rejection.

Major comments:

1. Although the HT-29 is allogeneic to the reconstituted human immune systems it might elicit specific alloreactive T cell responses. This is also suggested by the *in vivo* killing assays. However, in order to demonstrate that the observed CD4+ T cell rejection *in vivo* is indeed alloantigen specific it should be shown that rejection of a HLA mismatched cell line would not be improved by primary HT-29 implantation and rejection. For the *in vitro* killing assays HLA-matched tumor cells sharing some MHC class II molecules with HT-29 should be used to define the alloantigen response in more detail that is responsible for the observed HT-29 rejection.

We thank the reviewer for the critical comments and suggestions.

To address whether CD4 T cell rejection of HT-29 *in vivo* is alloantigen specific and better define the potential alloantigen response, we exploited the use of: 1) CA46, a B cell lymphoma cell line expressing fully mismatched HLAs to HT-29, and 2) HCT116, another colorectal tumor cell line expressing mostly mismatched HLA-II except for DPA1. The HLA information of HT-29, CA46, and HCT116 is provided in the new Supplementary Table I. In mice that had previously rejected HT-29 tumors, CA46 implantation led to a heterogeneous tumor outcome, where some mice developed large tumors quickly, and others had lower tumor burden compared to the control mice that never experienced HT-

29 tumors (Supplementary Fig. 3). This result suggests that primary HT-29 rejection provided little or partial protection against CA46 challenge *in vivo*. It is possible that some HT-29-recognizing TCRs cross-react to CA46 allo-antigens, as it has been reported that CD4 TCRs can promiscuously recognize different HLA-II alleles with shared epitopes (Penzotti, Doherty et al. 1996, Ou, Mitchell et al. 1997, Doherty, Penzotti et al. 1998).

Notably, primary HT-29 rejection resulted in robust eradication of HCT116 tumors, whereas mice never experienced HT-29 tumors were not able to mount the protection (Supplementary Fig. 3). This result supports the possibility that CD4 T cells reject HT-29 tumors through DPA1 allo-recognition

These new data are now provided in Supplementary Fig. 3 and discussed in lines 168-181 and 402-408. The additional mice used for this experiment have been included in the updated pie chart of Fig 1A and the corresponding figure legend. The total number of mice and donors used in this manuscript have been updated in lines 106-108.

2. The authors define at least three cytotoxic CD4+ T cell subsets in the microenvironment of regressing tumors. Which of the identified CD4+ T cell subsets degranulates primarily in response to HT-29 restimulation *in vitro*? This could be characterized by co-staining of markers that were identified by single cell RNA sequencing and CD107a as a degranulation marker after HT-29 restimulation *in vitro*.

To characterize the CD4 T cell subset that degranulates in response to HT-29 stimulation *in vitro*, we stained CD4 T cells in the killing assay with PD1, IL7R and CD107a. Most of the CD4 T cells were PD1⁺IL7R⁻, and majority of PD1⁺IL7R⁻ cells were CD107a⁺. These data can be found in the new Supplementary Fig. 7E and the description of it is in lines 323-326. The results suggest that the degranulating CD4 T cells are similar to the C1-PRF1 population identified by scRNAseq.

3. The authors claim that cytotoxicity of CD4+ T cells is both necessary and sufficient for the observed tumor cell rejection. In its current version, the manuscript does not support these claims. They have not shown that adoptive transfer of CD4+ T cells alone controls HT-29 growth in the absence of other immune compartments, nor that the cytotoxicity of these CD4+ T cells is required for HT-29 rejection. In order to substantiate these claims the authors should block TNF *in vivo*, render CD4+ T cells perforin deficient prior to adoptive transfer into HT-29 carrying mice without any other human immune system reconstitution.

We thank the reviewer for the comment and have modified our claim to make it more accurate. CD4 T cells are necessary for tumor control *in vivo*, as shown by the *in vivo* T cell depletion experiments. Further, these cells are also necessary and sufficient to kill HT-29 tumor cells *ex vivo*, although we cannot rule out contributions from other cell types *in vivo* (lines 343-346).

Although adoptive transfer can address certain questions, we believe an *in vitro* system allows us to interrogate mechanisms of tumor killing by T cells more directly, as it circumvents other obstacles such as how CD4 CTLs survive, migrate and infiltrate tumors as a differentiated population.

To address whether cytotoxicity is required for HT-29 killing, we rendered CD4 T cells perforin defective by treating them with a well-established inhibitor of the perforin pathway, concanamycin A (CMA) (Cachot, Bilous et al. 2021). We observed that inhibition of perforin with CMA resulted in reduced killing of tumor cells *in vitro* by CD4 T cells. These new data suggest that cytotoxicity function is required for tumor killing by CD4 T cells and is now discussed in lines 326-328, 346-347, and 578-580, and new figure 6E.

Minor comments:

1. Interferon gamma should be consistently abbreviated in the same fashion throughout the manuscript (IFNG instead of INFG).

We have corrected the typo and made sure that interferon gamma is consistently abbreviated as IFNG.

2. The authors argue in their discussion that the presented model might be used to identify tumor antigens. However, since the observed CD4+ T cell response is most likely alloreactive, this claim should be toned down unless an autologous tumor to reconstituted human immune system model can be established.

We thank the reviewer for the comment and have toned down the claims in lines 402-408 as suggested by the reviewer.

REFERENCE

- Amadori, A., R. Zamarchi, G. De Silvestro, G. Forza, G. Cavatton, G. A. Danieli, M. Clementi and L. Chieco-Bianchi (1995). "Genetic control of the CD4/CD8 T-cell ratio in humans." Nat Med **1**(12): 1279-1283.
- Cachot, A., M. Bilous, Y. C. Liu, X. Li, M. Saillard, M. Cenerenti, G. A. Rockinger, T. Wyss, P. Guillaume, J. Schmidt, R. Genolet, G. Ercolano, M. P. Protti, W. Reith, K. Ioannidou, L. de Leval, J. A. Trapani, G. Coukos, A. Harari, D. E. Speiser, A. Mathis, D. Gfeller, H. Altug, P. Romero and C. Jandus (2021). "Tumor-specific cytolytic CD4 T cells mediate immunity against human cancer." Sci Adv **7**(9).
- Caushi, J. X., J. Zhang, Z. Ji, A. Vaghasia, B. Zhang, E. H. Hsiue, B. J. Mog, W. Hou, S. Justesen, R. Blosser, A. Tam, V. Anagnostou, T. R. Cottrell, H. Guo, H. Y. Chan, D.

Singh, S. Thapa, A. G. Dykema, P. Burman, B. Choudhury, L. Aparicio, L. S. Cheung, M. Lanis, Z. Belcaid, M. El Asmar, P. B. Illei, R. Wang, J. Meyers, K. Schuebel, A. Gupta, A. Skaist, S. Wheelan, J. Naidoo, K. A. Marrone, M. Brock, J. Ha, E. L. Bush, B. J. Park, M. Bott, D. R. Jones, J. E. Reuss, V. E. Velculescu, J. E. Chaft, K. W. Kinzler, S. Zhou, B. Vogelstein, J. M. Taube, M. D. Hellmann, J. R. Brahmer, T. Merghoub, P. M. Forde, S. Yegnasubramanian, H. Ji, D. M. Pardoll and K. N. Smith (2021). "Transcriptional programs of neoantigen-specific TIL in anti-PD-1-treated lung cancers." Nature **596**(7870): 126-132.

Cohen, M., A. Giladi, O. Barboy, P. Hamon, B. Li, M. Zada, A. Gurevich-Shapiro, C. G. Beccaria, E. David, B. B. Maier, M. Buckup, I. Kamer, A. Deczkowska, J. Le Berichel, J. Bar, M. Iannacone, A. Tanay, M. Merad and I. Amit (2022). "The interaction of CD4(+) helper T cells with dendritic cells shapes the tumor microenvironment and immune checkpoint blockade response." Nat Cancer **3**(3): 303-317.

Doherty, D. G., J. E. Penzotti, D. M. Koelle, W. W. Kwok, T. P. Lybrand, S. Masewicz and G. T. Nepom (1998). "Structural basis of specificity and degeneracy of T cell recognition: pluriallelic restriction of T cell responses to a peptide antigen involves both specific and promiscuous interactions between the T cell receptor, peptide, and HLA-DR." J Immunol **161**(7): 3527-3535.

Gu-Trantien, C., S. Loi, S. Garaud, C. Equeter, M. Libin, A. de Wind, M. Ravoet, H. Le Buanec, C. Sibille, G. Manfouo-Foutsop, I. Veys, B. Haibe-Kains, S. K. Singhal, S. Michiels, F. Rothe, R. Salgado, H. Duvillier, M. Ignatiadis, C. Desmedt, D. Bron, D. Larsimont, M. Piccart, C. Sotiriou and K. Willard-Gallo (2013). "CD4(+) follicular helper T cell infiltration predicts breast cancer survival." J Clin Invest **123**(7): 2873-2892.

Li, J. P., C. Y. Wu, M. Y. Chen, S. X. Liu, S. M. Yan, Y. F. Kang, C. Sun, J. R. Grandis, M. S. Zeng and Q. Zhong (2021). "PD-1(+)CXCR5(-)CD4(+) Th-CXCL13 cell subset drives B cells into tertiary lymphoid structures of nasopharyngeal carcinoma." J Immunother Cancer **9**(7).

Oh, D. Y., S. S. Kwek, S. S. Raju, T. Li, E. McCarthy, E. Chow, D. Aran, A. Ilano, C. S. Pai, C. Rancan, K. Allaire, A. Burra, Y. Sun, M. H. Spitzer, S. Mangul, S. Porten, M. V. Meng, T. W. Friedlander, C. J. Ye and L. Fong (2020). "Intratumoral CD4(+) T Cells Mediate Anti-tumor Cytotoxicity in Human Bladder Cancer." Cell **181**(7): 1612-1625 e1613.

Okoye, A., H. Park, M. Rohankhedkar, L. Coyne-Johnson, R. Lum, J. M. Walker, S. L. Planer, A. W. Legasse, A. W. Sylwester, M. Piatak, Jr., J. D. Lifson, D. L. Sodora, F. Villinger, M. K. Axthelm, J. E. Schmitz and L. J. Picker (2009). "Profound CD4+/CCR5+ T cell expansion is induced by CD8+ lymphocyte depletion but does not account for accelerated SIV pathogenesis." J Exp Med **206**(7): 1575-1588.

Ou, D., L. A. Mitchell and A. J. Tingle (1997). "HLA-DR restrictive supertypes dominate promiscuous T cell recognition: association of multiple HLA-DR molecules with susceptibility to autoimmune diseases." J Rheumatol **24**(2): 253-261.

Penzotti, J. E., D. Doherty, T. P. Lybrand and G. T. Nepom (1996). "A structural model for TCR recognition of the HLA class II shared epitope sequence implicated in susceptibility to rheumatoid arthritis." J Autoimmun **9**(2): 287-293.

Petitprez, F., A. de Reynies, E. Z. Keung, T. W. Chen, C. M. Sun, J. Calderaro, Y. M. Jeng, L. P. Hsiao, L. Lacroix, A. Bougouin, M. Moreira, G. Lacroix, I. Natario, J. Adam, C. Lucchesi, Y. H. Laizet, M. Toulmonde, M. A. Burgess, V. Bolejack, D. Reinke, K. M. Wani, W. L. Wang, A. J. Lazar, C. L. Roland, J. A. Wargo, A. Italiano, C. Sautes-Fridman, H. A. Tawbi and W. H. Fridman (2020). "B cells are associated with survival and immunotherapy response in sarcoma." Nature **577**(7791): 556-560.

Stelekati, E., H. Shin, T. A. Doering, D. V. Dolfi, C. G. Ziegler, D. P. Beiting, L. Dawson, J. Liboon, D. Wolski, M. A. Ali, P. D. Katsikis, H. Shen, D. S. Roos, W. N. Haining, G. M. Lauer and E. J. Wherry (2014). "Bystander chronic infection negatively impacts development of CD8(+) T cell memory." Immunity **40**(5): 801-813.

Thommen, D. S., V. H. Koelzer, P. Herzog, A. Roller, M. Trefny, S. Dimeloe, A. Kiialainen, J. Hanhart, C. Schill, C. Hess, S. Savic Prince, M. Wiese, D. Lardinois, P. C. Ho, C. Klein, V. Karanikas, K. D. Mertz, T. N. Schumacher and A. Zippelius (2018). "A transcriptionally and functionally distinct PD-1(+) CD8(+) T cell pool with predictive potential in non-small-cell lung cancer treated with PD-1 blockade." Nat Med **24**(7): 994-1004.

Veatch, J. R., S. M. Lee, C. Shasha, N. Singhi, J. L. Szeto, A. S. Moshiri, T. S. Kim, K. Smythe, P. Kong, M. Fitzgibbon, B. Jesernig, S. Bhatia, S. S. Tykodi, E. T. Hall, D. R. Byrd, J. A. Thompson, V. G. Pillarisetty, T. Duhon, A. McGarry Houghton, E. Newell, R. Gottardo and S. R. Riddell (2022). "Neoantigen-specific CD4(+) T cells in human

melanoma have diverse differentiation states and correlate with CD8(+) T cell, macrophage, and B cell function." Cancer Cell **40**(4): 393-409 e399.

Wei, S. C., J. H. Levine, A. P. Cogdill, Y. Zhao, N. A. S. Anang, M. C. Andrews, P. Sharma, J. Wang, J. A. Wargo, D. Pe'er and J. P. Allison (2017). "Distinct Cellular Mechanisms Underlie Anti-CTLA-4 and Anti-PD-1 Checkpoint Blockade." Cell **170**(6): 1120-1133 e1117.

Workel, H. H., J. M. Lubbers, R. Arnold, T. M. Prins, P. van der Vlies, K. de Lange, T. Bosse, I. C. van Gool, F. A. Eggink, M. C. A. Wouters, F. L. Komdeur, E. C. van der Slikke, C. L. Creutzberg, A. Kol, A. Plat, M. Glaire, D. N. Church, H. W. Nijman and M. de Bruyn (2019). "A Transcriptionally Distinct CXCL13(+)CD103(+)CD8(+) T-cell Population Is Associated with B-cell Recruitment and Neoantigen Load in Human Cancer." Cancer Immunol Res **7**(5): 784-796.

Wu, T., Y. Ji, E. A. Moseman, H. C. Xu, M. Manglani, M. Kirby, S. M. Anderson, R. Handon, E. Kenyon, A. Elkhoulou, W. Wu, P. A. Lang, L. Gattinoni, D. B. McGavern and P. L. Schwartzberg (2016). "The TCF1-Bcl6 axis counteracts type I interferon to repress exhaustion and maintain T cell stemness." Sci Immunol **1**(6).

Zheng, L., S. Qin, W. Si, A. Wang, B. Xing, R. Gao, X. Ren, L. Wang, X. Wu, J. Zhang, N. Wu, N. Zhang, H. Zheng, H. Ouyang, K. Chen, Z. Bu, X. Hu, J. Ji and Z. Zhang (2021). "Pan-cancer single-cell landscape of tumor-infiltrating T cells." Science **374**(6574): abe6474.